# A Factorized Low-Rank RNN Framework for Uncovering Independent Neural Latent Dynamics and Connectivity

Chengrui Li [* 1]  Yunmiao Wang [* 2]  Yule Wang [1]  Weihan Li [1]  Dieter Jaeger [2]  Anqi Wu [1]

## Abstract

Low-rank recurrent neural networks (lrRNNs) are a class of models that uncover low-dimensional latent dynamics underlying neural population activity. Although their functional connectivity is low-rank, it lacks independence interpretations, making it difficult to assign distinct computational roles to different latent dimensions. To address this, we propose the Factored Recurrent Neural Network (FacRNN), a generative lrRNN framework that assumes group-wise independence among latent dynamics while allowing flexible within-group entanglement. These independent latent groups allow latent dynamics to evolve separately, but are internally rich for complex computation. We reformulate the lrRNN under a variational autoencoder (VAE) framework, enabling us to introduce a partial correlation penalty that encourages independence between groups of latent dimensions. Experiments on synthetic, monkey M1, and mouse voltage imaging data show that FacRNN consistently improves the disentanglement and interpretability of learned neural latent trajectories in low-dimensional space and low-rank connectivity over baseline lrRNNs that do not encourage group-wise independence.

## 1. Introduction

Understanding neural dynamics and connectivity from high-dimensional recordings is a central challenge in neuroscience. Existing approaches typically fall into two categories: (1) methods that extract low-dimensional latent

dynamics or representations from neural population activity (Yu et al., 2005; Wu et al., 2017; Pandarinath et al., 2018; She & Wu, 2020; Aoi et al., 2020); and (2) models that operate directly in the high-dimensional neural space, enabling inference of connectivity structures (Pillow et al., 2008; Linderman et al., 2016; Roudi et al., 2015; Li et al., 2023). While the former provide insights into neural computation, they offer limited access to interpretable connectivity. The latter, meanwhile, do not directly estimate low-dimensional structure, which often leads to suboptimal discovery of the underlying neural dynamics.

Low-rank recurrent neural networks (lrRNNs) (Mastrogiuseppe & Ostojic, 2018; Beiran et al., 2021; Dubreuil et al., 2022; Schuessler et al., 2020a;b; Valente et al., 2022a) offer a promising middle ground, capturing both structured connectivity and low-dimensional dynamics by constraining the recurrent weights to a low-rank form. Recent methods, such as LINT (Valente et al., 2022b) and other related approaches (Pals et al., 2024), leverage the singular value decomposition (SVD) of low-rank connectivity matrices to extract interpretable latent subspaces. In these models, the left singular vectors define the latent dynamics, enabling both predictive modeling and functional interpretation.

However, SVD yields orthogonal components, which are not necessarily independent. In many experimental settings, it is desirable to identify independent latent subspaces that evolve separately. For example, in a motor decision-making task, Mante et al. (2013) demonstrated that latent trajectories under different task contexts (e.g., color vs. shape) are structured separately, suggesting that neural computations may arise from multiple independent sources.

To address this, techniques such as independent component analysis (ICA) (Hyvärinen & Oja, 2000) and disentangled variational autoencoders (VAEs) (Chen et al., 2018; Kim & Mnih, 2018) have been used to enforce independence. However, these approaches have two key limitations: (1) they are not dynamical models and therefore do not infer connectivity or recurrent dynamics; (2) they typically assume dimension-wise independence in the latent space, so that each latent dimension evolves independently, which is often too restrictive. For example, visual neurons have been shown to represent information in a factorized manner,

---

[*]Equal contribution Chengrui Li proposed the method, conducted the experiments, and wrote the manuscript. Yunmiao Wang performed the voltage-imaging experiments and provided the data for that section. [1]School of Computational Science & Engineering, Georgia Institute of Technology, Atlanta, USA [2]Department of Biology, Emory University, Atlanta, USA. Correspondence to: Chengrui Li <cnlichengrui@gatech.edu>.

*Proceedings of the 43rd International Conference on Machine Learning*, Seoul, South Korea. PMLR 306, 2026. Copyright 2026 by the author(s).

encoding features such as texture and contrast separately (Lee et al., 2025). While contrast might be able to be represented in a one-dimensional latent subspace, texture is inherently more complex and requires a higher-dimensional neural representation for accurate encoding. These reasons highlight that merely applying ICA or disentangled VAE to the neural data may not be accurate and insightful enough for disentangling task-relevant neural subspaces.

To address these limitations, we introduce the **Factored Recurrent Neural Network (FacRNN)**—a low-rank RNN framework that captures both neural connectivity and recurrent dynamics while enforcing factorizations between groups of latent dimensions (i.e., several latent subspaces). This approach offers three key benefits:
• It is a generative RNN model that enforces group-wise independence among latent dynamics, while allowing flexible interactions within each group.
• It yields interpretable low-rank sub-connectivities associated with each latent group, which can be viewed as distinct neural sub-circuits driving independent sources of task-related neural signals.
• Built on the VAE framework, it is flexible and extensible to complex RNN architectures.

## 2. Background

**Low-rank RNN.** lrRNN assumes that the observed neural sequence $\boldsymbol{x}(t) \in \mathbb{R}^N$ from $N$ neurons evolves with

$$\frac{\mathrm{d}\boldsymbol{x}(t)}{\mathrm{d}t} = -\boldsymbol{x}(t) + \boldsymbol{W} \int_0^\infty \psi(\tau)\sigma(\boldsymbol{x}(t-\tau))\,\mathrm{d}\tau \\ + \boldsymbol{b} + \boldsymbol{U}\boldsymbol{\eta}(t) + \boldsymbol{\epsilon}(t), \tag{1}$$

where $\boldsymbol{W} \in \mathbb{R}^{N \times N}$ is the low-rank connectivity matrix, $\boldsymbol{b} \in \mathbb{R}^N$ is the neuron background intensity vector, $\psi(\tau)$ is a history convolution kernel (with $\int_0^\infty \psi(\tau)\,\mathrm{d}\tau = 1$), $\sigma$ is a nonlinear activation function (e.g., $\tanh(\cdot)$), $\boldsymbol{\eta}(t) \in \mathbb{R}^D$ denotes external inputs (e.g., stimuli or behavioral variables), $\boldsymbol{U} \in \mathbb{R}^{N \times D}$ is a learnable linear map from external input space to neuron space, and $\boldsymbol{\epsilon}(t) \in \mathbb{R}^N$ are i.i.d. noise samples (e.g., Gaussian).

In practice, learning such a stochastic differential equation usually relies on its discretized version

$$\boldsymbol{x}^{(t)} | \boldsymbol{x}^{(t-1)}, \dots, \boldsymbol{x}^{(t-L)} = \boldsymbol{W} \sum_{l=1}^L \psi_l \, \sigma\big(\boldsymbol{x}^{(t-l)}\big) \\ + \boldsymbol{b} + \boldsymbol{U}\boldsymbol{\eta}^{(t)} + \boldsymbol{\epsilon}^{(t)}, \tag{2}$$

where $\{\boldsymbol{x}^{(t)}\}_{t=1}^T$ is the discretized neural sequence in $T$ time bins. $\boldsymbol{\psi} \in \mathbb{R}_{\geqslant 0}^L$ is the history convolution kernel (with $\sum_{l=1}^L \psi_l = 1$).

To obtain latent dynamics $\boldsymbol{z}(t)$ given the rank-$K$ weight matrix $\boldsymbol{W}$, LINT (Valente et al., 2022b) parameterized $\boldsymbol{W} = \boldsymbol{A}\boldsymbol{B}$, where $\boldsymbol{A} \in \mathbb{R}^{N \times K}$ is the left singular matrix, and $\boldsymbol{B} \in \mathbb{R}^{K \times N}$ is the transposed and singular-value-scaled right singular matrix. $\boldsymbol{x}(t)$ is projected to $\boldsymbol{z}(t)$ via $\boldsymbol{B}$.

**VAE for low-dimensional latent.** A variational auto-encoder (VAE) (Kingma, 2013) with linear encoder $q(\boldsymbol{z}|\boldsymbol{x})$ and linear decoder $p(\boldsymbol{x}|\boldsymbol{z};)$ can be viewed as a dimensionality reduction tool similar to probabilistic principal component analysis (PPCA), used to find the low-dimensional neural latent $\boldsymbol{z} \in \mathbb{R}^K$. The prior $p(\boldsymbol{z})$ is typically chosen as a standard normal prior.

To fit this VAE model, we optimize the standard evidence lower bound (ELBO) (Blei et al., 2017):

$$\frac{1}{T} \sum_{t=1}^T \mathrm{ELBO} = \frac{1}{T} \sum_{t=1}^T \mathbb{E}_{q(\boldsymbol{z}^{(t)}|\boldsymbol{x}^{(t)})} \left[ \ln p\big(\boldsymbol{x}^{(t)}\big|\boldsymbol{z}^{(t)}\big) \right] \\ - \mathrm{KL}\big(q\big(\boldsymbol{z}^{(t)}\big|\boldsymbol{x}^{(t)}\big)\|p\big(\boldsymbol{z}^{(t)}\big)\big). \tag{3}$$

However, the inferred latent components $z_1, \dots, z_K$ are generally entangled, and any invertible affine transformation $\boldsymbol{P} \in \mathbb{R}^{K \times K}$ can form an equivalent solution $\boldsymbol{P}^{-1}\boldsymbol{z}$. To see this, assume the encoder is $\boldsymbol{z} = \boldsymbol{B}\boldsymbol{x} + \boldsymbol{d}$ and the decoder is $\boldsymbol{x} = \boldsymbol{A}\boldsymbol{z} + \boldsymbol{c}$. Without noise for simplicity, $\boldsymbol{x} = \boldsymbol{A}\boldsymbol{z} + \boldsymbol{c} = \boldsymbol{A}(\boldsymbol{B}\boldsymbol{x} + \boldsymbol{d}) + \boldsymbol{c} = \boldsymbol{A}\boldsymbol{P}(\boldsymbol{P}^{-1}\boldsymbol{B}\boldsymbol{x} + \boldsymbol{P}^{-1}\boldsymbol{d}) + \boldsymbol{c}$, so the transformed $\boldsymbol{z}' = \boldsymbol{P}^{-1}\boldsymbol{B}\boldsymbol{x} + \boldsymbol{P}^{-1}\boldsymbol{d} = \boldsymbol{P}^{-1}\boldsymbol{z}$ is the equivalent latent under the encoder $(\boldsymbol{P}^{-1}\boldsymbol{B}, \boldsymbol{P}^{-1}\boldsymbol{d})$ and decoder $(\boldsymbol{A}\boldsymbol{P}, \boldsymbol{c})$.

## 3. Factored recurrent neural network (FacRNN)

### 3.1. Reformulate low-rank RNN using VAE

**Reformulation.** Learning an independent latent structure for lrRNN is not as simple as learning an orthogonal latent structure, which can be obtained via specific decompositions such as SVD. To address this, we first reformulate the lrRNN using the aforementioned VAE framework by adding nonlinearity and a history convolution $\boldsymbol{\psi} \in \mathbb{R}_{\geqslant 0}^L$ with $\sum_{l=1}^L \psi_l = 1$ to the original linear encoder:

$$\boldsymbol{z}^{(t)} | \boldsymbol{x}^{(t-1)}, \dots, \boldsymbol{x}^{(t-L)} = \boldsymbol{B} \sum_{l=1}^L \psi_l \, \sigma\big(\boldsymbol{x}^{(t-l)}\big) + \boldsymbol{d} + \boldsymbol{\epsilon}_{\mathrm{enc}}^{(t)}, \tag{4}$$

where $\boldsymbol{B} \in \mathbb{R}^{K \times N}$ and $\boldsymbol{d} \in \mathbb{R}^K$ are the encoder parameters, and $\boldsymbol{\epsilon}_{\mathrm{enc}}^{(t)}$ are i.i.d. encoder noise samples. We then write the decoder as

$$\boldsymbol{x}^{(t)} | \boldsymbol{z}^{(t)} = \boldsymbol{A}\boldsymbol{z}^{(t)} + \boldsymbol{c} + \boldsymbol{U}\boldsymbol{\eta}^{(t)} + \boldsymbol{\epsilon}_{\mathrm{dec}}^{(t)}, \tag{5}$$

where $\boldsymbol{A} \in \mathbb{R}^{N \times K}$ and $\boldsymbol{c} \in \mathbb{R}^N$ are the decoder parameters, $\boldsymbol{U} \in \mathbb{R}^{N \times D}$ maps external inputs $\boldsymbol{\eta}^{(t)} \in \mathbb{R}^D$ into the $N$-dimensional observation space, and $\boldsymbol{\epsilon}_{\text{dec}}^{(t)}$ are i.i.d. decoder noises. The term $\boldsymbol{U}\boldsymbol{\eta}^{(t)}$ separates task-driven exogenous input from internally generated recurrence through $\boldsymbol{AB} \sum_l \psi_l \sigma(\cdot)$. When no external inputs are available, $\boldsymbol{\eta}^{(t)}$ can be omitted or set to zero. Combining encoder and decoder, we obtain a generative lrRNN form for $\boldsymbol{x}^{(t)}$:

$$
\begin{aligned}
\boldsymbol{x}^{(t)} \big| \boldsymbol{x}^{(t-1)}, \ldots, \boldsymbol{x}^{(t-L)} &= \boldsymbol{AB} \sum_{l=1}^{L} \psi_l \, \sigma\big(\boldsymbol{x}^{(t-l)}\big) \\
&+ (\boldsymbol{Ad} + \boldsymbol{c}) + \boldsymbol{U}\boldsymbol{\eta}^{(t)} + \big(\boldsymbol{A}\boldsymbol{\epsilon}_{\text{enc}}^{(t)} + \boldsymbol{\epsilon}_{\text{dec}}^{(t)}\big).
\end{aligned} \tag{6}
$$

This can be viewed as an $L$-th order lrRNN ($L$ history time steps dependency), where the rank-$K$ connectivity is $\boldsymbol{W} := \boldsymbol{AB} \in \mathbb{R}^{N \times N}$ and the background intensity vector is $\boldsymbol{b} := (\boldsymbol{Ad} + \boldsymbol{c}) \in \mathbb{R}^N$.

**Latent dynamics.** A key benefit of expressing lrRNN in the VAE framework is that it allows deriving latent dynamics by replacing $\{\boldsymbol{x}^{(t-L)}, \ldots, \boldsymbol{x}^{(t-1)}\}$ in Eq. (4) with the form in Eq. (5):

$$
\begin{aligned}
\boldsymbol{z}^{(t)} \big| \boldsymbol{z}^{(t-1)}, \ldots, \boldsymbol{z}^{(t-L)} &= \boldsymbol{B} \sum_{l=1}^{L} \psi_l \\
\sigma\Big(\boldsymbol{A}\boldsymbol{z}^{(t-l)} &+ \boldsymbol{c} + \boldsymbol{U}\boldsymbol{\eta}^{(t-l)} + \boldsymbol{\epsilon}_{\text{dec}}^{(t-l)}\Big) + \boldsymbol{d} + \boldsymbol{\epsilon}_{\text{enc}}^{(t)}.
\end{aligned} \tag{7}
$$

In the special case when $\sigma$ is the identity function, the latent dynamics simplify to

$$
\begin{aligned}
\boldsymbol{z}^{(t)} =& \boldsymbol{BA} \sum_{l=1}^{L} \psi_l \boldsymbol{z}^{(t-l)} + (\boldsymbol{Bc} + \boldsymbol{d}) + \boldsymbol{B} \sum_{l=1}^{L} \psi_l \boldsymbol{U}\boldsymbol{\eta}^{(t-l)} \\
&+ \left( \boldsymbol{B} \sum_{l=1}^{L} \psi_l \boldsymbol{\epsilon}_{\text{dec}}^{(t-l)} + \boldsymbol{\epsilon}_{\text{enc}}^{(t)} \right),
\end{aligned} \tag{8}
$$

where $\boldsymbol{J} := \boldsymbol{BA} \in \mathbb{R}^{K \times K}$ denotes the corresponding latent recurrent connectivity. Although Eqs. (2)–(8) allow a general history length $L$, all experiments use $L = 1$ (equivalently $\psi_1 = 1$) for simplicity, which is sufficient to demonstrate the benefits of FacRNN. The framework can be easily extended to $L > 1$ when needed.

### 3.2. Factored recurrent neural network (FacRNN)

**Definition of factorization.** Another benefit of formulating lrRNN under the VAE framework is that the low-dimensional latent variable $\boldsymbol{z}^{(t)}$ from the encoder distribution is naturally amenable to factorization. We assume latent $\boldsymbol{z} \in \mathbb{R}^K$ is factorized by $G$ groups, each with

internal group rank $H_g$, satisfying $K = H_1 + H_2 + \cdots + H_G$. For simplicity, we denote the $g$-th group as $\boldsymbol{z}_g = (z_{g,1}, z_{g,2}, \ldots, z_{g,H_g}) \in \mathbb{R}^{H_g}, \ \forall g \in \{1, ..., G\}$, so that $\boldsymbol{z} = (\boldsymbol{z}_1, \ldots, \boldsymbol{z}_G)$. Then, the factorization is defined as $\perp\!\!\!\perp_{g=1}^{G} \boldsymbol{z}_g \iff$

$$
p(\boldsymbol{z}) = \prod_{g=1}^{G} p(\boldsymbol{z}_g), \quad p(\boldsymbol{z}_g) \neq p(z_{g,1}) \cdots p(z_{g,H_g}), \tag{9}
$$

$\forall g \in \{1, \ldots, G\}$. Intuitively, this relaxes dimension-wise independence (as in ICA or $\beta$-TCVAE): each group may form a multi-dimensional subspace with rich within-group coupling, while different groups should not share statistical dependence. See Fig. 1(b).

**Inference.** We introduce two inference approaches: one tailored for linear dynamics (Eq. (8)); while the other is flexible for general non-linear dynamics (Eq. (7)).

*Inference via block-diagonal latent structure.* To require group-wise independence for the linear case in Eq. (8), a straightforward way is to constrain the latent recurrence matrix $\boldsymbol{J} = \boldsymbol{BA}$ to be block-diagonal, so that latent groups do not interact with each other. If latent components within a group are entangled with each other (non-separable), the block will be similar to a Jordan matrix, i.e., the block cannot be diagonalized (see Apendix. A.1 for more details). Therefore, we can either: (1) freely learn $\boldsymbol{A}$ and $\boldsymbol{J}$ and solve a least-squares problem $\boldsymbol{J} = \boldsymbol{BA}$ to recover $\boldsymbol{B}$; or (2) jointly learn $\boldsymbol{A}$ and $\boldsymbol{B}$ while penalizing the off-block-diagonal elements of $\boldsymbol{J} = \boldsymbol{BA}$ to promote group-wise independence. We call this approach **block-diagonal RNN (bdRNN)**.

*Inference via partial correlation.* For the nonlinear case in Eq. (7), there is no $\boldsymbol{J}$ matrix due to the nonlinear $\sigma$ function. Thus, we directly deal with Eq. (9). Following (Li et al., 2025), we achieve group-wise independence by optimizing the target function

$$
\mathcal{L} = \frac{1}{T} \sum_{t=1}^{T} \text{ELBO}\left(\boldsymbol{x}^{(t)}\right) - \beta \cdot \text{KL}\left(q(\boldsymbol{z}) \middle\| \prod_{g=1}^{G} q(\boldsymbol{z}_g)\right). \tag{10}
$$

The second term in Eq. (10) is the partial correlation (PC), where the aggregated posterior $q(\boldsymbol{z}) = \frac{1}{T} \sum_{t=1}^{T} q(\boldsymbol{z}, \boldsymbol{x}^{(t)}) = \sum_{t=1}^{T} q(\boldsymbol{z} | \boldsymbol{x}^{(t)}) q(\boldsymbol{x}^{(t)})$ is defined as in (Makhzani et al., 2015). Since each data point is equally contributed, $q(\boldsymbol{x}^{(t)}) = \frac{1}{T}$ and hence $q(\boldsymbol{z})$ can be viewed as a Gaussian kernel density estimation over $\{\boldsymbol{z}^{(t)}\}_{t=1}^{T}$ in latent space. In practice, the PC term in the form KL divergence is numerically estimated by all $\boldsymbol{x}^{(t)}$ in each sequence. PC penalizes dependency between latent groups: when $q(\boldsymbol{z}) = \prod_{g=1}^{G} q(\boldsymbol{z}_g)$, PC =

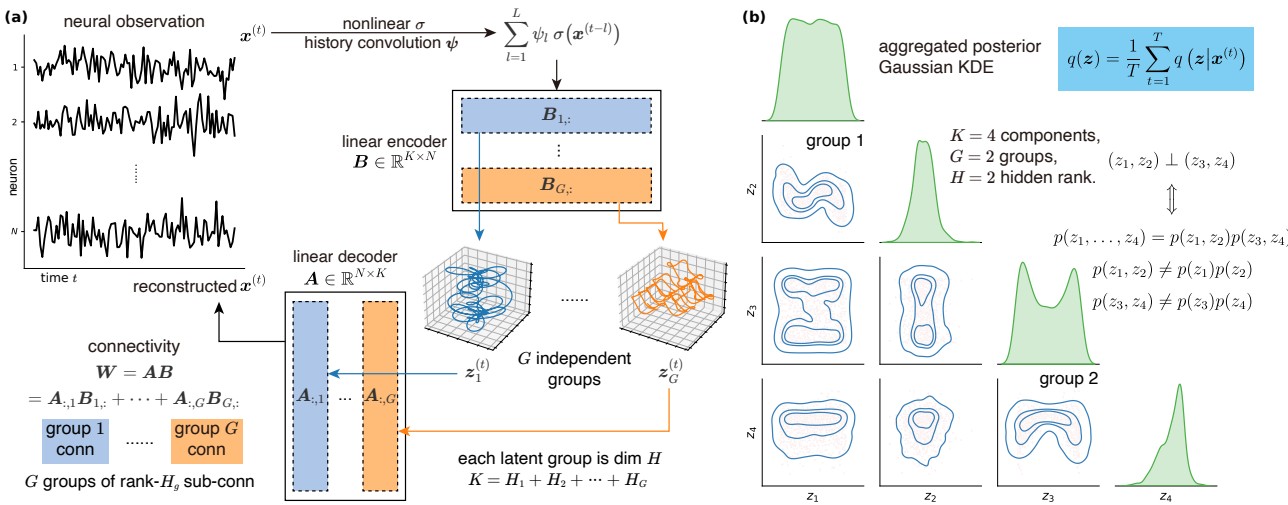

*Figure 1.* **(a)**: Schematic of the FacRNN, showing independent latent groups and the corresponding low-rank connectivities. **(b)**: Group-wise independence: $(z_1, z_2) \perp (z_3, z_4)$, while within-groups are highly entangled, and components from different groups are marginally independent.

$\mathrm{KL}\left(q(\boldsymbol{z}) \middle\| \prod_{g=1}^{G} q(\boldsymbol{z}_g)\right) = 0$; otherwise, PC $> 0$ and the penalty is scaled by a hyperparameter $\beta > 0$. We call this approach **Factored RNN (FacRNN)**.

**Sub-circuit connectivity.** With FacRNN, we can obtain a group-wise independent latent $\boldsymbol{z}$, where each group $g$ has its group-rank $H_g$. Specifically, we write

$$\boldsymbol{x}^{(t)} = \boldsymbol{A}\boldsymbol{z}^{(t)} + \boldsymbol{c} + \boldsymbol{U}\boldsymbol{\eta}^{(t)} + \boldsymbol{\epsilon}^{(t)}$$
$$= [\boldsymbol{A}_{:,1}, \ldots, \boldsymbol{A}_{:,G}]\left[z_1^{(t)}, \ldots, z_G^{(t)}\right]^{\mathrm{T}} \quad (11)$$
$$+ \boldsymbol{c} + \boldsymbol{U}\boldsymbol{\eta}^{(t)} + \boldsymbol{\epsilon}^{(t)},$$

where $\boldsymbol{A}_{:,g}$ is the $H_g$-dimensional embedding of the $g$-th latent group subspace in the observational space $\mathbb{R}^N$. Accordingly,

$$\boldsymbol{W} = \boldsymbol{A}\boldsymbol{B} = \sum_{g=1}^{G} \boldsymbol{A}_{:,g}\boldsymbol{B}_{g,:} =: \sum_{g=1}^{G} \boldsymbol{W}_g, \quad (12)$$

where $\boldsymbol{W}_g$ is the rank $H_g$ sub-connectivity associated with the $g$-th group. Each $\boldsymbol{W}_g$ represents a sub-circuit within the neural population that encodes an independent source of neural activity. With task-related labels for stimuli or choices, for example, we can identify which sub-circuits encode stimulus features and which encode choice signals. This distinction is essential for understanding how different neural circuits support perception, decision-making, and the functional organization of the brain. Fig. 1(a) is a complete schematic of the overall framework.

**Benefits.** There are three main benefits of FacRNN.
• First, the PC term in FacRNN in fact encompasses RNNs

with no factorization when $G = 1$ (then PC $\equiv 0$ by definition, so the objective reduces to the standard lrRNN/VAE objective) and full component-wise independence when $G = K$ (as in ICA (Hyvärinen & Oja, 2000), $\beta$-TCVAE (Chen et al., 2018), and FactorVAE (Kim & Mnih, 2018), although this is often very restrictive).
• Second, unlike bdRNN, by setting a sufficiently large group-rank $H_g$, FacRNN can automatically detect the effective group rank and $H'_g$ and introduce dummy dimensions $H_g - H'_g$. (Appendix A.3.1).
• Third, FacRNN is compatible with both linear (Eq. (8)) and nonlinear (Eq. (7)) dynamics, and naturally extends to more general nonlinear dynamical systems of the form $\boldsymbol{x}^{(t)} = f(\boldsymbol{x}^{(t-1)}, \ldots, \boldsymbol{x}^{(t-L)}) + \boldsymbol{\epsilon}^{(t)}$. Even in the absence of explicit connectivity in such a general nonlinear form, FacRNN still enables learning a factorized low-dimensional latent representation with a nonlinear encoder. Therefore, FacRNN serves as a more general inference framework than bdRNN.

## 4. Experiments

**Methods for comparison.** We compare RNN-based methods that emphasize different latent structure representations, without involving their underlying sampling or inference strategies.
• **lrRNN**: The standard low-rank RNN model in Eq. (2).
• **LINT** (Valente et al., 2022b): lrRNN with connectivity matrix parameterized by SVD.
• **bdRNN**: Our block-diagonal RNN model, which enforces $\boldsymbol{BA}$ to be block-diagonal and is theoretically valid only when $\sigma$ is the identity function.
• **FacRNN**: Our FacRNN model in Eq. (10), which achieves

latent factorization by penalizing the PC term.

## 4.1. Synthetic dataset

**Dataset.** To generate the latent, we simulate Lorenz and Thomas' cyclically symmetric dynamics with $\Delta t = 0.1$ for 2000 steps using RK4 (Dormand & Prince, 1980). Fig. 5 in Appendix A.3.1 visualizes the latent plots. To simulate $\boldsymbol{x}^{(t)} \in \mathbb{R}^{20}$, we need to make sure the dataset also satisfies the recurrent relationships in Eq. (6) and Eq. (7). Since $\boldsymbol{z}^{(t)}$ follows the generative process in Eq. (7), we can get the parameters including $\boldsymbol{A}, \boldsymbol{B}$, and hence generate the observed data $\boldsymbol{x}^{(t)}$ using the fitted $\boldsymbol{A}$ and $\boldsymbol{B}$ with i.i.d. Gaussian external inputs $\boldsymbol{\eta}^{(t)}$ (with $D = N$ and $\boldsymbol{U} = \boldsymbol{I}$ in simulation) and random Gaussian noises. Check the code in our public code repository for all the details.

**Experimental setup.** Since the ground truth contains two independent latent groups of rank 3, we fit bdRNN and FacRNN with $(G, H) = (2, 3)$. To conduct a more comprehensive experiment, we also include two additional intermediate methods: (1) **lrRNN+ICA** that performs a post-hoc ICA on the latent estimated from lrRNN; and (2) **FacRNN-full** that runs FacRNN with $(G, H) = (6, 1)$ (i.e., full latent disentanglement via FacRNN). All methods are trained for 5000 epochs using the Adam optimizer (Kingma, 2014) with a learning rate of $10^{-3}$ and a batch size of 128. The ablation study in Appendix A.3.1 cross-validates $\beta = 20$ and demonstrates the flexibility of $(G, H)$ setting as the second benefit of FacRNN. All methods are run 10 times with different random seeds.

**Latent evaluations.** To evaluate the estimated latent unsupervisedly, we compute the PC of the estimated latent on the test set to check whether different methods uncover desired group structures. Fig. 2(a) shows that FacRNN achieves the lowest PC ($\approx 0.13$), on the same order as the test set ground-truth latent PC ($\approx 0.1$), whereas unfactorized lrRNN remains much higher ($\approx 0.86$), indicating a successful recovery of group-wise independent latent dynamics. Comparing FacRNN-full and lrRNN+ICA, we see that end-to-end training in FacRNN-full can lead to better latent factorization. In contrast, the latent space of lrRNN, without any independence constraints, may not be factorizable, especially when the VAE structure contains nonlinearities, which makes post-hoc ICA unreliable or even non-decomposable. In other words, the factorization objective in an end-to-end model influences both the encoder and decoder during training, whereas imposing factorization post hoc after learning the latent representation is ineffective.

Given we know the true latent dynamics $\left\{ \boldsymbol{z}'^{(t)} \in \mathbb{R}^K \right\}_{t=1}^{T}$ in this synthetic dataset, we can align the estimated latent dynamics to the ground truth. In general, if we have $G$ groups,

we can match the estimated latent groups $\boldsymbol{z}_1^{(t)}, \ldots, \boldsymbol{z}_G^{(t)}$ to the true latent groups $\boldsymbol{z}'^{(t)}_1, \ldots, \boldsymbol{z}'^{(t)}_G$, as illustrated in Fig. 7 in Appendix. Specifically, we create an $\mathbf{R2} \in (-\infty, 1]^{G \times G}$ matrix where $\mathrm{R2}_{g_1, g_2}$ is the $R^2$ score of aligning the estimated latent $\boldsymbol{z}_{g_2}^{(t)}$ to the true latent $(\boldsymbol{z}')_{g_1}^{(t)}$ via an affine transformation, by solving a least squares problem. Then, the best match is obtained by finding a mutually exclusive assignment from true groups $g'$ to estimated groups $g$ that maximizes the total $R^2$ score. This is essentially a linear sum assignment problem in graph theory (Crouse, 2016). After the assignment, we report the average $R^2$ over the finally matched pairs.

**Results.** Although different methods reconstruct similar observation sequences (all methods are approximately 0.85 reconstruction $R^2$ through their learned recurrent weights), their uncovered latent dynamics differ from each other in terms of factorized structure. The latent $R^2$ in Fig. 2(a) shows that FacRNN recovers the latent dynamics with the highest accuracy, while standard lrRNN and LINT perform noticeably worse. Among all methods, only FacRNN faithfully recovers the group-wise independent structure present in the ground truth. Fig. 2(b) visualizes the estimated latent dynamics, confirming that FacRNN's latent trajectories are better aligned with the ground truth than others. To further explore the estimated latent from methods without group-wise independence assumption, we perform another post-hoc analysis in Fig. 9 in Appendix A.3.1, demonstrating that partitioning latent components from methods without the assumption of group-wise structure has combinatorially high complexity, which is a practical limitation for the baseline methods.

In terms of parameter estimation, although all methods yield similar estimates of the overall recurrent connectivity $\boldsymbol{W} = \boldsymbol{AB}$ (with connectivity correlations around 0.85), they find different $\boldsymbol{A}$ and $\boldsymbol{B}$. Particularly, the sub-connectivities discovered by FacRNN match the ground truth the best (Fig. 2(a) and (c)).

## 4.2. Monkey M1 data

**Dataset.** We use neural spike train recordings from the macaque M1 cortex (Gallego et al., 2020). The dataset consists of firing rate data from 168 trials, 14 time bins, and 154 neurons, recorded while the animal performed a center-out reaching task with eight movement directions.

**Experimental setup.** Since the ground-truth latent structure is unknown, we explore a range of model configurations based on $K = 2$ and $K = 4$ latent dimensions and analyze their outcomes.
- For $K = 2$, we learn lrRNN and LINT. We also learn a FacRNN with two independent rank-1 latent groups

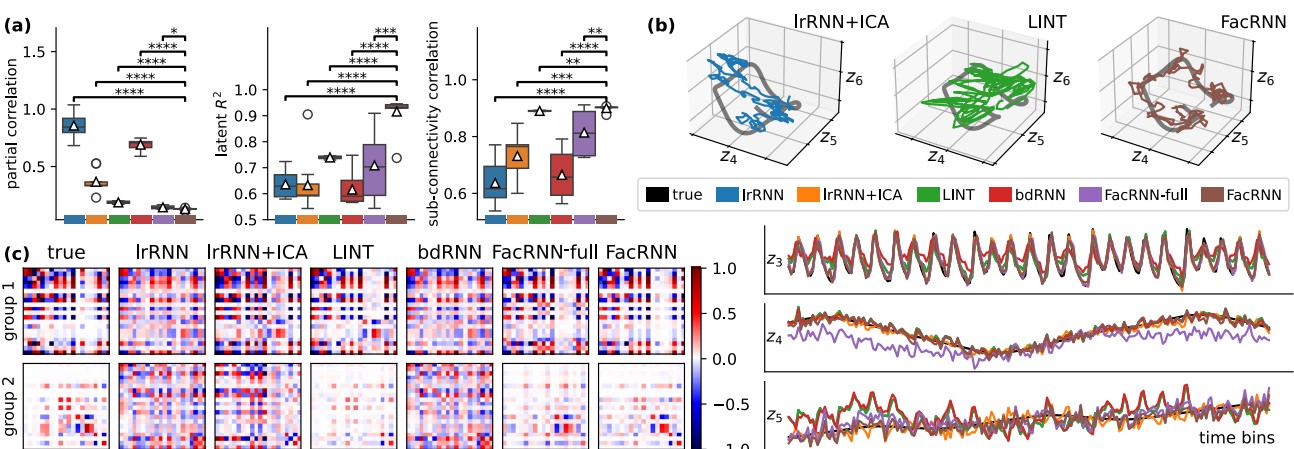

*Figure 2.* **(a)**: The PC and $R^2$ of the estimated latent, and the connectivity correlation. The starbars indicate the pairwise $t$-test significance levels. Arrows indicate the higher or lower the better. **(b)**: Group 2 latent trajectories with the true dynamics in 3D plots; and the 1D trajectories from different methods on selected latent components. All latent dimensions are plotted in Fig. 8 in Appendix A.3.1. **(c)**: The learned sub-connectivities from different methods and the ground truth.

$(G, H) = (2, 1)$. Additionally, since our FacRNN supports arbitrary encoder and decoder architectures as illustrated in the inference section, we also try a FacRNN model with an MLP encoder and decoder, denoted as FacRNN-MLP.

• For $K = 4$, we again learn lrRNN and LINT. For FacRNN, we learn two independent rank-2 latent groups $(G, H) = (2, 2)$. We also run SMC (Pals et al., 2024) as an ancillary reference (similar center-out M1 task in their work, but SMC has a different inference pipeline from ours).

The training procedures are similar to the synthetic experiments (see our code for details). After training, we attempt to align the learned latent groups to $x$-coordinates (horizontal) and $y$-coordinates (vertical) of the monkey's hand movement trajectories (the true trajectories in Fig. 3(a)).

**Results.** For $K = 2$, lrRNN and LINT perform poorly in aligning to the $x$ and $y$ movement separately (Fig. 3). FacRNN improves the alignment score from about 0.5 to 0.65. However, the trajectories remain visually poor, suggesting that rank-1 latent dynamics are not expressive enough to capture the M1 neural activities responsible for the reaching task.

With more components $K = 4$ in a larger latent space, lr-RNN and LINT achieve improved alignment, but still lack any form of latent factorization. Although SMC is a more advanced method than LINT due to its sophisticated inference approach, their reported gains in the original paper relied on using direction labels. In our setting, where such labels are not provided, the sophisticated inference alone makes SMC perform even worse than LINT. In contrast, FacRNN with two rank-2 latent groups achieves an alignment $R^2$ score of about 0.8. The trajectories to all eight directions are significantly better separated than all other

configurations. We also tried FacRNN with 4 rank-1 groups. Due to its rigid full independence assumption, it fails to group components meaningfully into $x$ and $y$-aligned latent subspaces, resulting in a lower alignment score. These results indicate that higher-than-rank-1 is important to model the dynamics of the $x$ or $y$ coordinates effectively.

To understand whether the poor factorization in $K = 2$ stems from the linear architecture of the encoder and decoder, we replace the linear encoder and decoder with more complicated MLP encoder and decoder, denoted as FacRNN-MLP. Its alignment $R^2$ remains lower than that of the FacRNN with two rank-2 groups. In particular, FacRNN-MLP fails to separate trajectories to $45°$ and $90°$ (colored by orange, green), and the same for $135°$ and $180°$ (colored by red and purple), which remain mixed. Besides, FacRNN-MLP does not support explicit connectivity estimation due to its nonlinear encoder and decoder (effective-connectivity proxies such as local Jacobian linearization are discussed in Appendix A.2). This further supports that higher-than-rank-1 latent groups are necessary for describing the $x$ or $y$ movement dynamics, and that simply having a more complicated VAE architecture might not resolve the factorization limitations of low-ranked groups.

We also compare the learned sub-connectivity matrices governing the dynamics for both $x$ and $y$ coordinates. Take the $K = 2$ FacRNN and the $K = 4$ FacRNN for example, both have two independent groups, but the $K = 4$ FacRNN has a higher-than-1 within-group rank (Fig. 3(b)). With factorization, the connectivity matrix can also be separated into two groups, one for the $x$ coordinate and one for the $y$ coordinate. Comparing the sub-connectivities between $K = 2$ and $K = 4$, they share some similar connections (indicated by the arcs between the stars in Fig. 3(b)). However, the rank-2

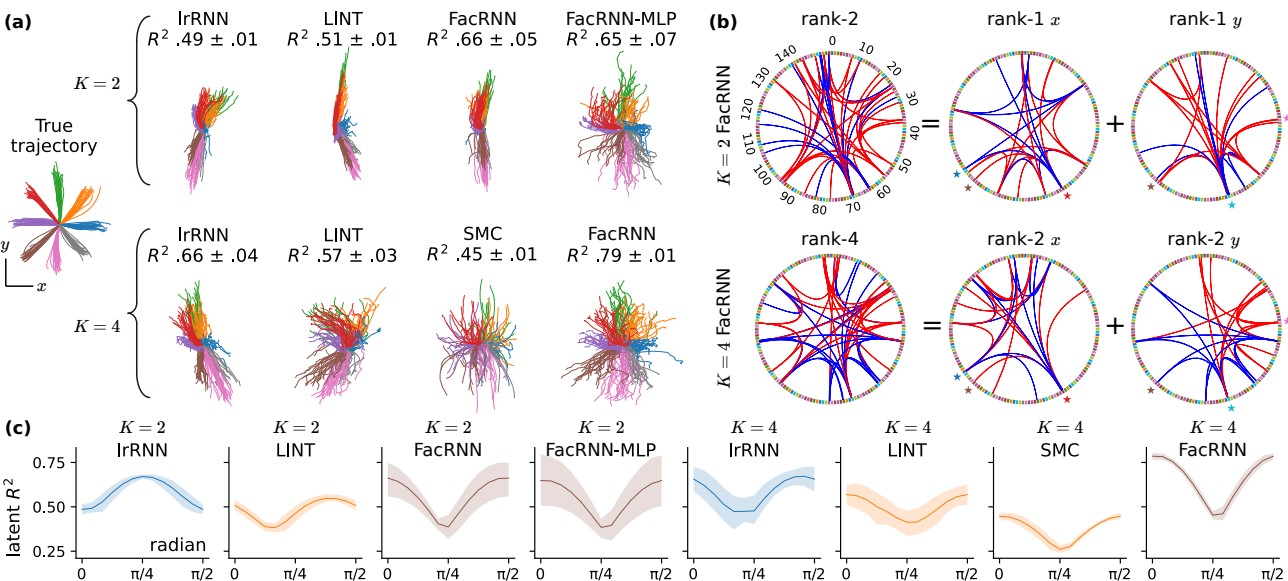

**Figure 3.** **(a)**: Recovered latent trajectories from different methods v.s. the ground-truth trajectories, and their alignment $R^2$ scores. **(b)**: The corresponding low-rank connectivity for $K = 2$ FacRNN and $K = 4$ FacRNN. For FacRNN, for example, the rank-4 connectivity can be decomposed into two rank-2 sub-connectivities, one responsible for the dynamics of the $x$-movement, and the other for the $y$-movement. **(c)**: The $R^2$ scores of aligning the estimated latent to rotated coordinate systems.

sub-connectivities are more expressive, which better support accurate approximation of the corresponding $x$ coordinate and $y$ coordinate latent dynamics. *This enables circuit-level hypotheses, such as one subnetwork supporting horizontal control and another vertical, and reveals how neural populations implement multiple variables through separable low-rank channels rather than a single entangled circuit.* This also helps explain why ICA-like post hoc separation may fail: factorization is coupled with encoder-decoder and connectivity learning during training, whereas a plain lr-RNN latent may already be too entangled for a subsequent linear rotation to recover meaningful groups. The learned connectivity matrices from all configurations are shown in Fig. 10 in Appendix A.3.2.

At the beginning of this experiment, we hypothesized that the $x$ and $y$ coordinates represent two underlying true latent groups. However, the eight directions in this experiment are rotationally symmetric, mathematically. This may cause the concern that we can align the factorized latent groups with respect to an arbitrarily rotated coordinate system (e.g., express the trajectory coordinate under the bases of direction $\frac{\pi}{4}$ and $\frac{3\pi}{4}$), instead of the horizontal/vertical one. To test this, we align the factorized latent to trajectories expressed in various rotated coordinate systems, ranging from 0 to $\frac{\pi}{2}$. If there is no factorization, the alignment score should be similar across different angles due to the rotational symmetry.

Fig. 3(c) indicates that for the three FacRNNs, the best alignment occurs under the canonical horizontal-vertical bases

(highest latent $R^2$ at 0 and $\frac{\pi}{2}$), and the alignment $R^2$ score drops significantly under the rotated bases, especially near $\frac{\pi}{4}$. In contrast, methods without factorization (standard lr-RNN and LINT) show flatter and more erratic alignment curves across different rotation angles, lacking a consistent preference for particular bases. These findings are biologically plausible, consistent with prior neurophysiological findings in primate motor cortex (Georgopoulos et al., 1982; Amirikian & Georgopoulos, 2003; Churchland et al., 2012), where neurons are not uniformly tuned to all directions, but more tuned along the cardinal axes, i.e., horizontal (left-right) and vertical (up-down), due to the body symmetry. Especially in these center-out reaching tasks, it is more natural for primates to understand and execute movements under the horizontal-vertical bases (i.e., the Cartesian coordinate system), rather than the polar coordinate system. *This reinforces the idea that M1 does not encode movement uniformly across directions, and certain axes may be overrepresented due to behavioral, biomechanical, habitual, or neural factors.* Taken together, these results support that the separation of $x$ and $y$ movement in our model is not merely an artifact of the model statistics, but may reflect a meaningful subspace underlying the neural data. Results from fitting the model to Poisson spike counts (Appendix A.3.2) similarly demonstrate that factorization enhances latent structure separation, yielding better alignment with hand movement trajectories.

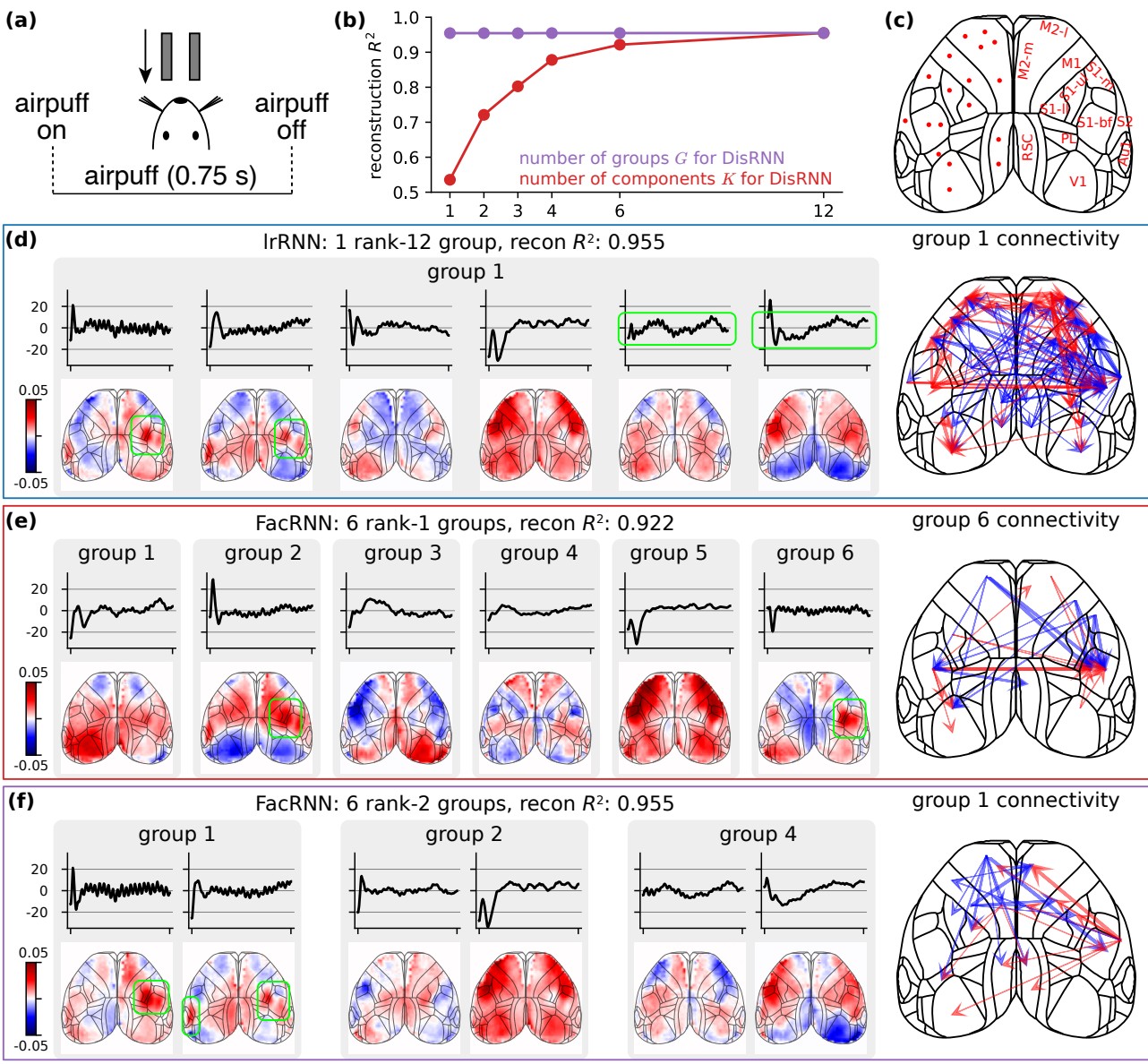

*Figure 4.* **(a)**: Experiment setup. **(b)**: Reconstruction performance w.r.t. different $(G, H)$ specifications with fixed $K = 12$, or different $K$ with $G = K$ and $H = 1$. **(c)**: Cortical map with region abbreviations. **(d)**: $(G, H) = (1, 12)$, i.e., standard lrRNN. Left: brain maps $\mathbf{A}_{:,g}$ and time series $\mathbf{z}_g^{(t)}$; right: rank-12 connectivity. Similar for **(e)** $(G, H) = (6, 1)$ and **(f)** $(G, H) = (6, 2)$.

## 4.3. Mouse dorsal cortex voltage imaging data

**Dataset and experimental setup.** The data set used in this study is a trial-averaged voltage imaging sequence from a mouse exposed to 30 Hz pulsed air-puff stimulus to the left whiskers in a directional lick task. The imaging protocol using the JEDI-1P voltage sensor followed previously published methods (Lu et al., 2023). The recording comprises 150 frames of $50 \times 50$ dorsal cortex voltage images during the left-side air puff lasting 0.75 seconds (Fig. 4(a)). Although FacRNN can in principle handle over-specified group ranks, overly large settings impair efficiency and model quality. Since the true latent structure is unknown,

fixing the total latent dimensionality $K$ while varying the group specification $(G, H)$ provides a controlled way to examine how different assumptions affect model performance and the latent interpretation. Thus, we fix $K = 12$ and investigate different numbers of groups $G \in \{1, 2, 3, 4, 6, 12\}$. When $G = 1$, there is no factorization and the model is lrRNN. Additionally, we explore fully factorized models by varying $K \in \{1, 2, 3, 4, 6, 12\}$ with $G = K$ and $H = 1$. The training procedures are similar to the previous experiments (see our code for details).

**Results.** Fig. 4(b) shows that increasing the total number of latent components in FacRNN improves reconstruction, indicating that a larger latent dimensionality preserves more information in the latent space. The improvement plateaus once the number of components reaches six, suggesting that a latent space of dimension six or higher is sufficient to capture the data structure. In comparison, fixing $K = 12$ maintains consistently high reconstruction performance across different group numbers. Varying the number of groups in this case does not affect observation reconstruction but leads to different degrees of latent factorization. This indicates that once sufficient latent capacity is available, the FacRNN framework allows us to explore and control the factorization structure via group specification—without compromising reconstruction accuracy.

To evaluate the latent, three configurations are selected and compared in Fig. 4(d-f), with a standard cortical region map and annotated ROIs (regions of interest) shown in Fig. 4(c). In each of Figs. 4(d-f), the left panel displays the brain map $\boldsymbol{A}_{:,g}$ and the corresponding time series $\boldsymbol{z}_g^{(t)}$ for each group $g$, while the right panel shows the associated group connectivity $\boldsymbol{W}_g$. In Fig. 4(d), learning a standard lrRNN without factorization results in mixed latent trajectories, where air-puff-related, stimulus-locked S1-bf (primary somatosensory-barrel field) oscillations (marked by the green squares) appear across multiple components. In contrast, FacRNN with 6 rank-1 groups $((G, H) = (6, 1)$, hence $K = 6$) in Fig. 4(e) isolates these stimulus-locked oscillations in the right S1-bf and right M2-m (secondary motor-medial), yielding a more localized representation. However, 6 components are less expressive compared to 12 in terms of reconstruction $R^2$. To address this, a FacRNN with 6 rank-2 groups $((G, H) = (6, 2)$, hence $K = 12$) in Fig. 4(f) is explored. In this configuration, air-puff-related oscillations locked to the air-puff stimulus are concentrated within group 1 and are not further separable. To quantify the degree of independence, total correlation (TC) is computed between and within groups. $(G, H) = (6, 1)$ yields an average between-group TC of 0.159, whereas $(G, H) = (6, 2)$ achieves a lower between-group TC of 0.110 and a within-group TC of 0.201, indicating better separation between groups while retaining within-group structure.

Regarding connectivity, Fig. 4(d) illustrates that without factorization, even low-rank connectivity remains hard to interpret. Under $(G, H) = (6, 1)$ (Fig. 4(e)), the rank-1 sub-connectivity linked to the stimulus-locked oscillatory latent (group 6) reveals strong bilateral connections between S1-bf regions and an influence from right M2 to right S1-bf. But with $(G, H) = (6, 2)$ (Fig. 4(f)), the rank-2 connectivity associated with the stimulus-locked oscillatory latent (group 1) offers more interesting interpretations, highlighting the contralateral barrel response with stimulus-locked oscillations that are shared here with M2 on the same side.

For example, it reveals a strong excitatory connection from S2 (secondary somatosensory) to M2-m. There are also connections from right S2 and left M2 to both sides of RSC (retrosplenial cortex), potentially indicating the formation of episodic memory of receiving the airpuff. Videos of group dynamics are provided in our public code repository.

## 5. Conclusion

In this work, we develop a factored low-rank RNN (FacRNN) framework that captures both the latent dynamics and connectivity structure of neural systems while relaxing the assumption of full independence among latent components. By combining the expressiveness of low-rank RNN (lrRNN) with a flexible variational inference approach, we enable group-wise independence of latent dynamics via a partial correlation penalty. Compared to lrRNN, fully factorized models, and SVD-based methods, FacRNN produces latent representations that better match known task variables and reveal an interpretable sub-connectivity structure. Our results across synthetic and real datasets suggest that group-wise independence is general and powerful for uncovering modular computation in brain activity.

Despite its advantages, FacRNN also has several limitations. First, the model requires pre-specifying both the number and the size of latent groups. Although our framework can automatically identify and suppress dummy or inactive components within each group, determining the appropriate group structure remains a hyperparameter selection challenge that may depend on domain knowledge or model selection criteria. Second, connectivity interpretability is currently limited to the case where both the encoder and decoder are linear. While this constraint enables direct estimation of functional connectivity between latent variables and observed neural activity, it represents a trade-off between interpretability and representational power. Extending the framework to allow for nonlinear encoders or decoders could improve modeling flexibility, but would make connectivity estimation less straightforward. Finally, the current real-world experiments do not incorporate external tasks or stimulus inputs, although doing so would be conceptually straightforward. Integrating such inputs offers an exciting direction for future work, potentially enabling the dissection of task-dependent latent dynamics and input-driven connectivity changes.

## Software and Data

Code to reproduce the experiments is available at https://github.com/JerrySoybean/facrnn.

## Acknowledgements

This work was supported by the Alfred P. Sloan Foundation, the NIH BRAIN Initiative U01NS131810 and R01 NS111470.

## Impact Statement

This paper presents work whose goal is to advance the field of Machine Learning and Computational Neuroscience. There are many potential societal consequences of our work, none of which we feel must be specifically highlighted here.

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

# A. Appendix

### A.1. Block-diagonal RNN (bdRNN)

Consider a first-order discrete linear system for simplicity,

$$\boldsymbol{z}^{(t+1)} = \boldsymbol{F}\boldsymbol{z}^{(t)}, \quad \boldsymbol{F} \in \mathbb{R}^{K \times K}. \tag{13}$$

Since not all matrices can be diagonalized, but all matrices have Jordan normal form, we can always write

$$\boldsymbol{F} = \boldsymbol{P}\boldsymbol{J}\boldsymbol{P}^{-1}, \tag{14}$$

where $\boldsymbol{J}$ is a Jordan normal form, consisting of several Jordan blocks on its block diagonal. If, without loss of generality, there are $G$ numbers of rank-$H$ latent groups evolving independently, then we have

$$\boldsymbol{J} = \begin{bmatrix} \boldsymbol{J}_1 & \boldsymbol{O} & \cdots & \boldsymbol{O} \\ \boldsymbol{O} & \boldsymbol{J}_2 & \cdots & \boldsymbol{O} \\ \vdots & \vdots & \ddots & \vdots \\ \boldsymbol{O} & \boldsymbol{O} & \cdots & \boldsymbol{J}_G \end{bmatrix}, \tag{15}$$

where

$$\boldsymbol{J}_g = \begin{bmatrix} \lambda_g & 1 & & & \\ & \lambda_g & 1 & & \\ & & \lambda_g & \ddots & \\ & & & \ddots & 1 \\ & & & & \lambda_g \end{bmatrix} \in \mathbb{R}^{H \times H}, \tag{16}$$

where $\lambda_g$ is the corresponding eigenvalue, so that within each group, latent components are entangled with each other, non-separable. Then,

$$\boldsymbol{z}^{(t)} = \boldsymbol{F}^t \boldsymbol{z}^{(0)} = \boldsymbol{P}\boldsymbol{J}^t \boldsymbol{P}^{-1}\boldsymbol{z}^{(0)}. \tag{17}$$

By defining $\boldsymbol{z}' = \boldsymbol{P}^{-1}\boldsymbol{z}$ (similar to the equivalence below Eq. (3)), $\boldsymbol{z}'^{(t)}$ evolves independently by groups, i.e.,

$$\boldsymbol{z}'_g(t) = \boldsymbol{J}^t_g \boldsymbol{z}'^{(0)}_g, \quad \boldsymbol{J}^t_g = \begin{bmatrix} \lambda^t_g & \binom{t}{1}\lambda^{t-1}_g & \binom{t}{2}\lambda^{t-2}_g & \cdots & \cdots & \binom{t}{H-1}\lambda^{t-H+1}_g \\ & \lambda^t_g & \binom{t}{1}\lambda^{t-1}_g & \cdots & \cdots & \binom{t}{H-2}\lambda^{t-H+2}_g \\ & & \ddots & \ddots & & \vdots \\ & & & \ddots & \ddots & \vdots \\ & & & & \lambda^t_g & \binom{t}{1}\lambda^{t-1}_g \\ & & & & & \lambda^t_g \end{bmatrix}. \tag{18}$$

### A.2. Nonlinear effective connectivity

FacRNN-MLP is included to test whether encoder/decoder expressiveness alone can substitute for explicit group-structured inductive bias, rather than to serve as the primary interpretable model. When the encoder and decoder are nonlinear, explicit low-rank connectivity $\boldsymbol{W} = \boldsymbol{A}\boldsymbol{B}$ is no longer available. Effective-connectivity proxies can instead be derived from local Jacobian-based linearization of the encoder–decoder map, and optionally complemented by attribution tools such as saliency maps or input×gradient analyses.

## A.3. Supplementary results

### A.3.1. SYNTHETIC

**Data generation.** Fig. 5 plots the dataset we created for our synthetic experiment.

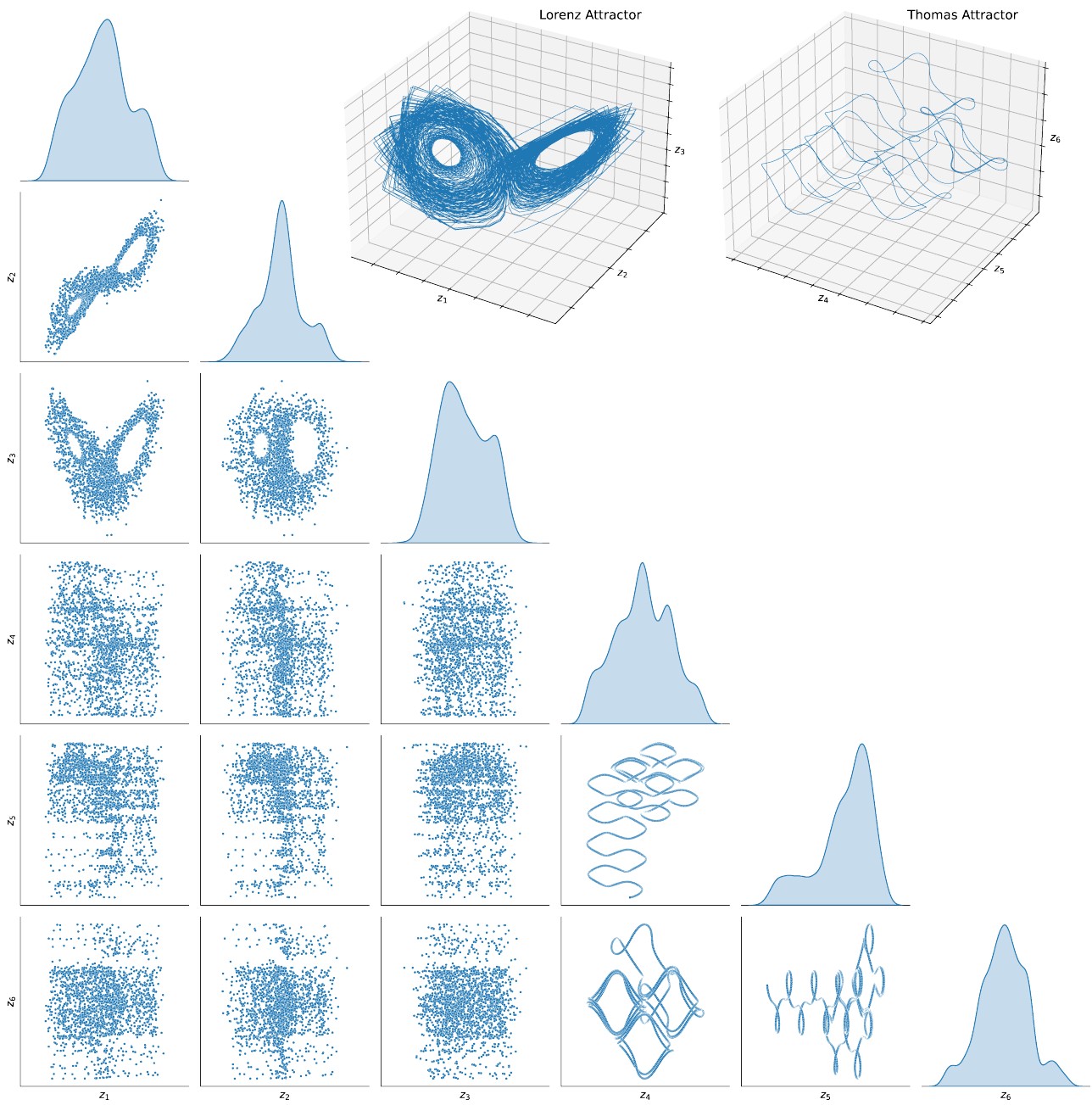

*Figure 5.* The synthetic latent consists of two independent groups.

**Ablation.**    To analyze the choice of the penalty coefficient $\beta$ of PC term in Eq. (10), we vary $\beta$ and plot the cross-validation results in Fig. 6. This supports our choice of $\beta = 20$ in our experiment that has good reconstruction, factorized latent estimation, and parameter estimation.

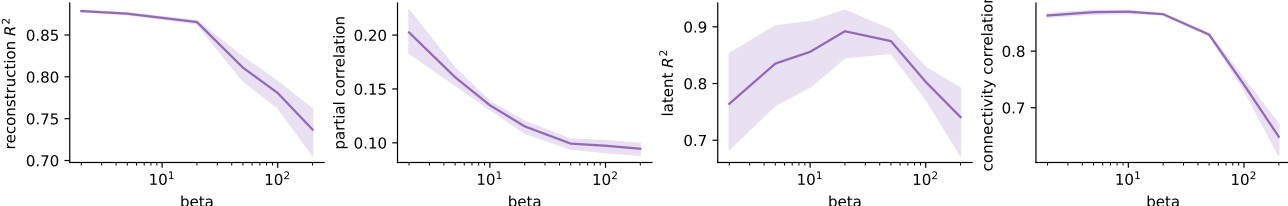

*Figure 6.* Metrics w.r.t. the PC penalty $\beta$ in FacRNN on the synthetic dataset.

To verify that FacRNN is flexible in setting the number of groups $G$ and group rank $H$, we run FacRNN with $(G, H) = (2, 4)$ and $(G, H) = (3, 4)$ against the true $(G, H) = (2, 3)$, and evaluate the latent alignment outcome. The latent $R^2$ for them are $0.84 \pm 0.14$ and $0.78 \pm 0.09$, respectively. All of them are better than other baseline methods, but worse than the correctly set FacRNN $((G, H) = (2, 3))$. Noticing that for both of them, the internal hidden rank reduces to the effective group rank of 3 automatically. And for $(G, H) = (3, 4)$, there is always a dummy group that aligns badly to both true groups. This validates the second benefit of FacRNN that it allows a flexibly set $(G, H)$ when the true structure is unknown.

For the over-specified $(G, H) = (2, 4)$ setting, we inspect effective rank within each learned group. PCA cumulative explained-variance ratios are $[0.643, 0.851, 0.99999, 1.000]$ for group 0 and $[0.653, 0.99943, 0.99993, 1.000]$ for group 1. Because a low-variance direction may still carry structured non-Gaussian information, we also apply normality tests to within-group PCA coordinates. The corresponding $p$-values are $[4.25 \times 10^{-10}, 1.35 \times 10^{-34}, 4.36 \times 10^{-17}, 0.774]$ for group 0 and $[2.25 \times 10^{-8}, 2.03 \times 10^{-4}, 1.63 \times 10^{-3}, 1.87 \times 10^{-2}]$ for group 1. Thus, the fourth coordinate of group 0 behaves like Gaussian dummy noise, while the first three coordinates in both groups remain clearly non-Gaussian, consistent with the true effective group rank $H = 3$ and with the ICA principle that independence is informative primarily for non-Gaussian sources (Hyvärinen & Oja, 2000).

**Alignment.**    Fig. 7 demonstrates the alignment procedure.

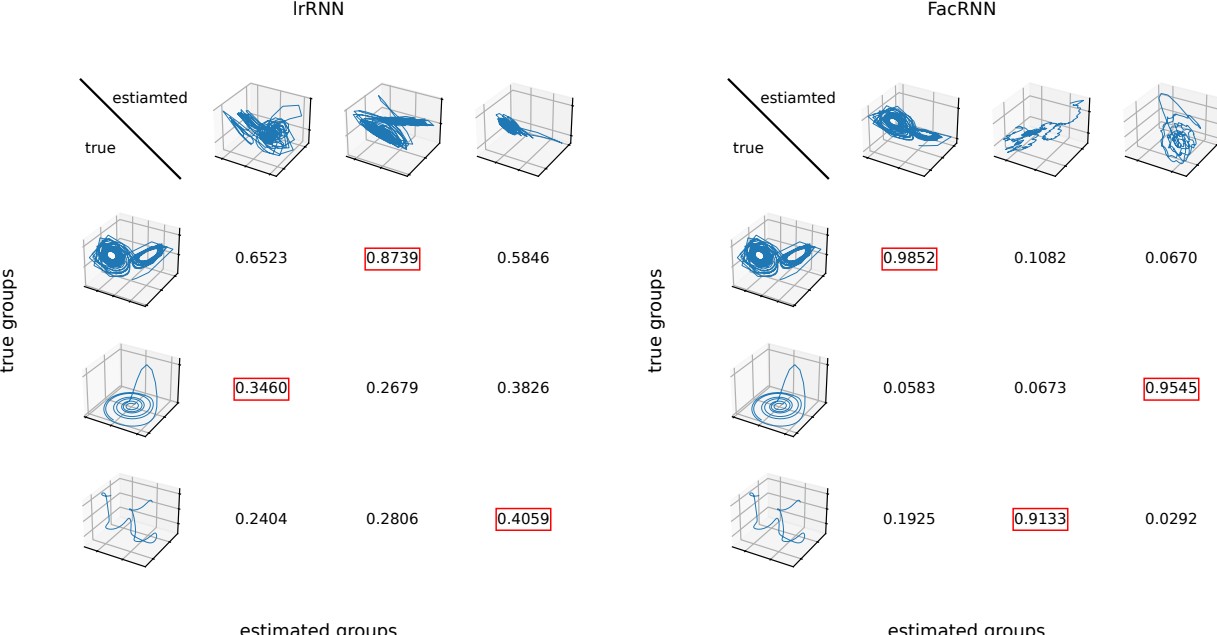

*Figure 7.* The latent dynamics alignment process. We try to align the estimated latent groups with the ground truth. The best match is marked by the red squared linear least-squares $R^2$ score.

**Latent dynamics.** Fig. 8 shows the estimated latent dynamics vs. the ground truth on all six latent components.

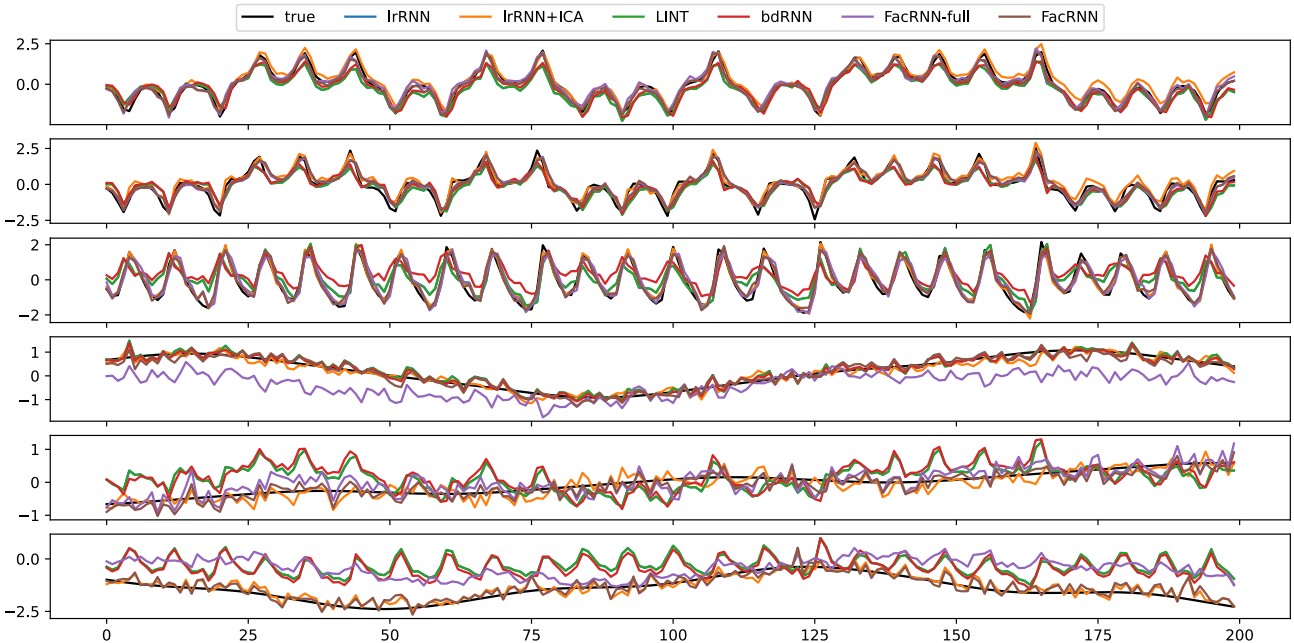

*Figure 8.* The estimated latent components by different methods on every latent component/dimension.

**All possible permutations.** For each method without the assumption of group-wise structure, we try all possible permutations of the latent components that partition the latent components into groups. In general, the complexity is

$$\frac{\prod_{g=1}^{G} \binom{(G+1-g)H}{H}}{G!} = \frac{K!}{G!(H!)^G}. \tag{19}$$

We instantiate this complexity in Fig. 9. Then, for each random seed, we also plot the latent alignment $R^2$ for all possible permutations, demonstrating that latent components are entangled with each other and in random orders. For some methods, the optimal permutation can achieve the same latent $R^2$ as FacRNN. However, this does not invalidate our FacRNN method, because FacRNN can explicitly factorize and assign components into groups. When the group specification is only moderately large, exhausting all permutations becomes intractable, which forms a theoretical limitation of baseline methods.

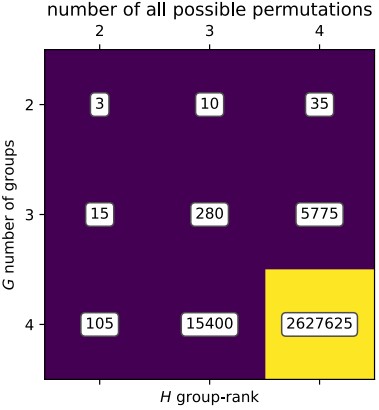 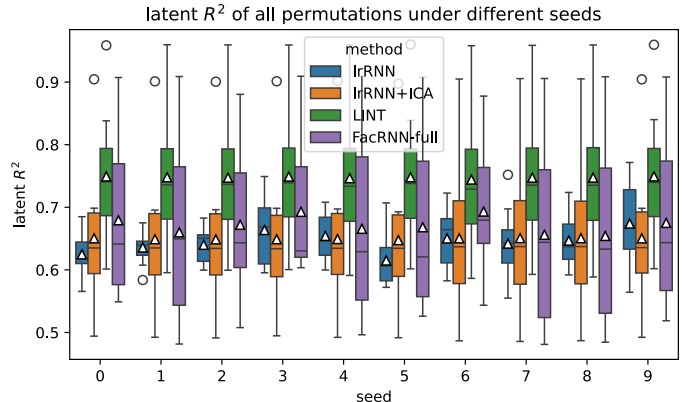

*Figure 9.* **Left**: Complexity demonstration matrix. **Right**: The latent $R^2$ of all possible permutations for methods without the assumption of group structure under different random seeds.

A.3.2. MONKEY M1

**Real-data $\beta$ selection.** On real data, where ground-truth latent $R^2$ is unavailable, we select $\beta$ using held-out reconstruction together with between-group PC. Table 1 shows this protocol on the monkey M1 dataset. The between-group PC has a clear elbow around $\beta = 10$–$20$, while the reconstruction log-likelihood remains nearly flat through the mid-range and decreases at very large $\beta$. This supports selecting a balanced operating range around $\beta = 10$–$20$, consistent with the synthetic ablation.

*Table 1.* Monkey M1 $\beta$ ablation. Values are mean (std) across five runs.

| $\beta$ | Recon. log-likelihood | Between-group PC |
|---|---|---|
| 1 | 380.001 (0.465) | 0.0165 (0.0075) |
| 2 | 379.923 (0.531) | 0.00922 (0.00447) |
| 5 | 379.740 (0.635) | 0.00237 (0.00180) |
| 10 | 379.622 (0.389) | 0.000932 (0.000701) |
| 20 | 379.661 (0.405) | 0.000620 (0.000383) |
| 50 | 379.642 (0.246) | 0.000427 (0.000195) |
| 100 | 379.668 (0.254) | 0.000302 (0.000115) |
| 200 | 379.422 (0.298) | 0.000220 (0.000159) |

**Connectivity matrices.** Fig. 10 shows the learned connectivity matrices from all methods.

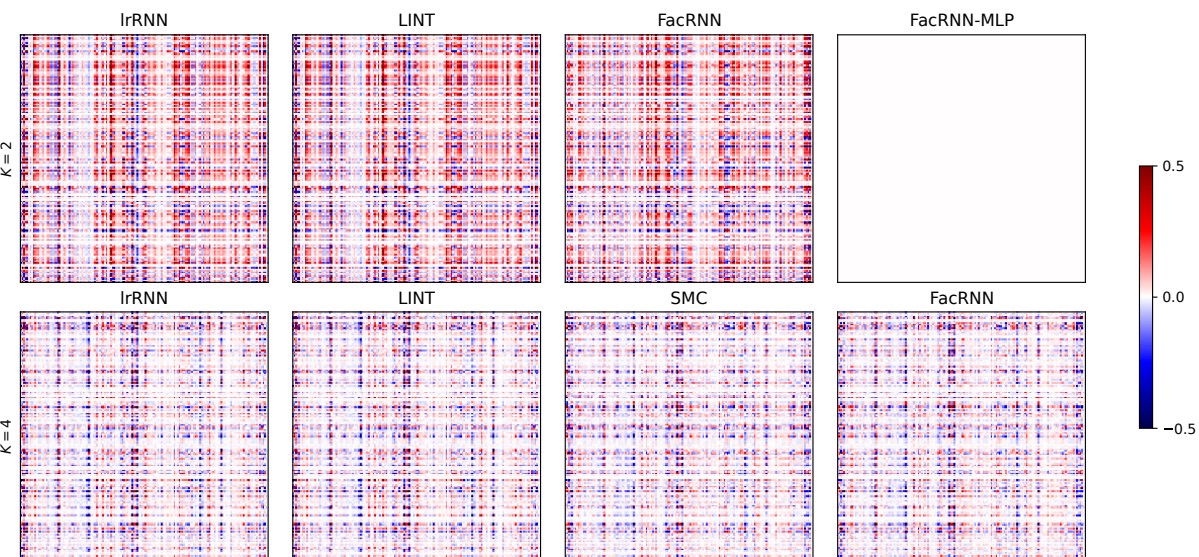

*Figure 10.* The learned connectivity matrices from all methods in Fig. 3(a), except for $K = 2$ FacRNN-MLP.

**Poisson spike count results.** To validate that our framework not only works for arbitrary VAE architectures but also works for different observation distributions, we fit different methods to the Poisson spike count data of the same dataset with $K = 2$. The latent alignment $R^2$ increases from plain lrRNN's $0.52 \pm 0.03$ to FacRNN's $0.76 \pm 0.02$ to FacRNN-MLP's $0.81 \pm 0.01$. This result is consistent with the result in the main content, confirming that factorization is an effective approach for obtaining biologically interpretable latent space structure.

