# OpenReview forum: "A Factorized Low-Rank RNN Framework for Uncovering Independent Neural Latent Dynamics and Connectivity"
_ICML.cc/2026/Conference — ICML 2026 spotlight_

### Official Review · Reviewer_7MBW · 2026-03-09

**Soundness:** 2
**Presentation:** 1
**Significance:** 2
**Originality:** 2
**Overall Recommendation:** 3
**Confidence:** 3

**Summary:**

This paper proposes a method for inferring factorized low-rank recurrent neural networks (RNNs) from data. The authors introduce a regularization term based on partial correlation (PC) into the ELBO objective, with the goal of encouraging group-wise independence among latent factors. The resulting model aims to recover interpretable latent sub-dynamics and associated connectivity structure. The method is first evaluated on a synthetic dataset to demonstrate its ability to recover structured latent dynamics. It is then applied to two neuroscience datasets, and it is compared with existing approaches for learning low-rank RNN models.

**Compliance With Llm Reviewing Policy:**

Affirmed.

**Final Justification:**

While I really appreciate the effort the authors have put into their response and their attempt to address my concerns, and I hope they find the review helpful in strengthening the paper, my concerns about the presentation remain. Since the paper’s originality appears to stem primarily from modifying the loss function, I believe that a paper of this type would require a particularly clear and careful presentation to warrant acceptance.

Moreover, although the additional non-Gaussianity analysis is helpful, I do not think it fully justifies the conclusion the authors draw. For Group 0, the distinction is relatively clear, since the normality-test p-values show a sharp separation between the first three coordinates and the fourth. However, for Group 1, the pattern appears much more gradual. The reported p-values decrease progressively, and the choice of a threshold such as 0.005 to classify the third coordinate as non-Gaussian while treating the fourth differently still seems somewhat arbitrary. As a result, I am not yet convinced that these results provide sufficiently strong evidence for the claimed “automatic detection” of the effective rank.

**Key Questions For Authors:**

1. The paper states that A and J are “freely learned,” after which B is estimated via least squares to obtain a block-diagonal structure. Could the authors clarify how this procedure leads to the block-diagonal structure in practice?

2. The role of the history filter is not entirely clear. It is difficult to determine from the main text or appendix whether it is actually used in the model. If it is not used, it might be clearer to present the canonical RNN formulation. If it is used, additional explanation of its purpose and implementation would be helpful.

3. In one experiment, R2 is computed after applying an affine transformation to evaluate whether the correct latent groups were recovered. In another experiment, however, the axes of the latent space are interpreted individually and linked to neuroscience interpretations. Since affine transformations include rotations, it would be helpful to clarify why axis alignment is important in the latter case but not the former.

4. The paper factorizes the latent activity but does not explicitly impose structure on the connectivity matrix. In this context, the statement explaining why ICA-like post-hoc separation may fail (“since the separation lies in the structured connectivity itself rather than only in latent activity.”) is not entirely clear. Could the authors elaborate on this point?

5. The paper states that (G=1) corresponds to a standard low-rank RNN, yet experiments using (K=6) and (G=1) appear to produce different results from a standard lrRNN (for the Mouse dorsal cortex voltage imaging data). Could the authors clarify the reason for this discrepancy?

**Limitations:**

see previous sections.

**Strengths And Weaknesses:**

The motivation for introducing a partial correlation regularization term to encourage independence among latent groups is technically reasonable.

The paper also evaluates the approach on both synthetic and real datasets, which helps illustrate the behavior of the method under controlled conditions as well as in practical applications.
Some aspects of the experimental evaluation could benefit from further clarification or consistency*. For example, the paper states that one experiment uses the same dataset as the SMC paper, but the cited datasets appear to differ, even though the tasks themselves are similar. This makes it difficult to directly interpret the comparison, especially when discussing the performance differences and the role of direction labels.

Relatedly, the choice of baseline methods varies across experiments. The paper motivates including SMC in only one experiment based on the assumption that the dataset matches the one used in the SMC work. If the datasets are in fact different, it would be helpful to provide a more systematic comparison across methods and experiments*.
While the factorization can be beneficial for interpretation in some domains* (e.g., certain sensory or motor systems), it may be less obvious for other brain areas such as the prefrontal cortex. A brief discussion acknowledging this could strengthen the paper.

First, the interpretation of sub-circuit connectivity (as illustrated in Figure 1) appears to rely on the partial correlation being exactly zero. This assumption could be clarified in the text.

Second, the relationship between the proposed method and existing approaches could be explained more precisely. For example, the proposed model appears equivalent to LINT when (G=1) and the partial correlation term is defined as zero. However, the paper states that “FacRNN in fact encompasses RNNs of no independence when (G=1).” In practice, this equivalence seems to hold only if the PC regularization term is explicitly set as zero, because when the value of PC is equal to zero (via training), this implies independence, not a standard lrRNN.

Additionally, the bdRNN formulation introduced in the paper appears to hold only in the linear case. Rather than presenting it as a full method alongside the others, it may be clearer to briefly introduce it and explain why the result is restricted to linear dynamics. Since experiments are already included, retaining them is still valuable; this suggestion mainly concerns presentation.
Finally, the paper states that the method can automatically detect the effective group rank. However, the appendix ablation study referenced for this claim does not explicitly report the effective group rank. Providing this information would help support the claim.

Regardless of all the problems I mentioned above, I can think of cases where this method can be beneficial, which makes the paper important.

The authors connected two different fields, inferring low-rank RNNs with factorized VAEs. Which is valuable and original.

---

> ### Author Rebuttal · Authors · 2026-03-29
>
> Dear Reviewer 7MBW,
>
> Thank you very much for your time and valuable comments on our paper. We highly appreciate your recognition of the strengths of our paper. Hopefully, the following responses could resolve most of your concerns and answer your questions.
>
> ### W1: Dataset inconsistency with SMC paper and varying baselines across experiments
>
> * We apologize for the omission. SMC was applied to a similar but different dataset. We included SMC because its original paper has a similar task.
> * However, SMC relies on an importance-sampling-based latent inference pipeline, rather than the VAE-based RNN family studied in this paper, so direct comparability is limited. To avoid confusion, we can remove SMC from the main text for clarity.
>
> ### W2: Clarify sub-circuit interpretation vs. exact PC=0 assumption
>
> * Our objective is group-wise independence, but in finite-sample settings, PC is not expected to be exactly zero. Even for the ground-truth latent, its PC is close to zero.
> * We use PC as a penalty to encourage group factorization, rather than requiring PC=0. As a sanity-check, since FacRNN recovers high-quality latents (e.g., high $R^2$), a PC magnitude (≈0.13 in the synthetic dataset) on the same order as the ground-truth close-to-zero PC (≈0.1), compared with naive lrRNN's large PC (≈0.86), is reasonable. We will add this to Fig. 2(a).
>
> ### W3: Relationship to LINT and the $G=1$ equivalence claim
>
> We would like to clarify that: **when $G=1$, PC=0 by definition**, so the objective reduces to the base lrRNN, with essentially no PC penalty since the left and right terms in the KL div are the same.
>
> ### W4: bdRNN presentation, linear-only restriction, and block-diagonal procedure
>
> We include bdRNN because it reflects a clean theoretical consequence for linear ODE under group-wise factorized recurrence. We acknowledge that bdRNN is an intermediate method rather than our main contribution. We can move the bdRNN discussion and results to Appendix A.1 for clarity.
>
> ### W5: Claim that FacRNN can automatically detect effective group rank is unsupported
>
> We have added the following experiment. The synthetic ground truth has group rank $H=3$. With an over-specified $H=4$, we run PCA within each learned group and inspect the cumulative explained variance ratio. For both groups, the curves plateau after 3 dims.
> ```
> Group 0 explained variance ratio: [0.64299828, 0.85100238, 0.99999126, 1.]
> Group 1 explained variance ratio: [0.65260285, 0.99942801, 0.99992943, 1.]
> ```
> This validates our claim that FacRNN can automatically detect dummy dimensions and reveal effective group rank. We will include these results in the next revision.
>
> ### Q1: How to learn block-diagonal in bdRNN?
>
> We freely learn block-diagonal $J$ and $A$, Since, we want $J=BA$, we then solve $J\approx BA$ for $B$ (least squares/pseudoinverse). This guarantees group decoupling in latent recurrence through $J$.
>
> ### Q2: Is the history filter actually used in the model?
>
> See the response to W2 and Q3 in Reviewer Thwq.
>
> ### Q3: Why does axis alignment matter in the neural experiments but not the synthetic ones?
>
> In both settings, we essentially do the same alignment: matching learned latent groups to a reference, simulated latents in synthetic data, or behavior/task variables in real data as an interpretable proxy. In synthetic experiments, estimated groups are aligned to true groups (up to affine transforms within groups). On monkey M1, we align learned groups to candidate behavioral coordinates, e.g., Cartesian $(x,y)$ or polar $(\rho,\theta)$. Quantitatively, $R^2$ is much higher for Cartesian ($≈0.79$) than for polar ($≈0.24$), indicating the learned factors track horizontal/vertical components more closely than radius/angle, supporting a neuroscience interpretation.
>
> ### Q4: Why does ICA-like post-hoc separation fail, the "structured connectivity" argument?
>
> * First, **low-rank** is already a structural constraint on connectivity. Beyond that, we do not have additional hard structural priors on the real datasets. This choice is orthogonal to the post-hoc ICA discussion.
> * We include lrRNN+ICA mainly as a negative control for a two-stage pipeline: fit a plain lrRNN, then apply ICA (can only find dim-wise/full factorization). So, we use it only as a baseline for FacRNN-full.
> * Our experiments show **end-to-end** FacRNN performs better, indicating that latent disentanglement is **coupled** with encoder-decoder learning. Otherwise, a plain lrRNN latent may be too entangled for meaningful subsequent factorization.
>
> ### Q5: Why does $G=1$ ($K=6$) differ from the standard lrRNN in the mouse data?
>
> * We apologize for the notation confusion in Fig. 4. "$G$ rank-$H$ groups" means "$G$ groups, each has within-group-rank $H$".
> * We clarify that when $G=1$, FacRNN theoretically reduces to a naive lrRNN without factorization as explained in W3. There is no separate "FacRNN with $G=1$" versus a distinct "lrRNN" baseline. Our results contain only one such configuration.

---

> > ### Author Rebuttal · Reviewer_7MBW · 2026-04-02
> >
> > I would like to thank the authors for their detailed and thoughtful responses. I appreciate the clarifications provided for W2, W3, Q1, Q4, and Q5, which have addressed my concerns.
> >
> > W1: Regardless of whether SMC is included in the main text, I believe the more fundamental issue lies in the inconsistency of baselines across experiments. Rigorous and consistent experimental comparisons are particularly important for this line of work, as they play a key role in guiding future research and ensuring fair evaluation of proposed methods.
> >
> > For W4, I agree with the authors’ perspective. It may improve clarity to briefly introduce the linear method in the main text, while deferring detailed explanations to the appendix.
> >
> > For W5, it appears from the results that the plateau for Group 1 occurs after two dimensions. Could the authors please clarify why their response refers to three dimensions?
> >
> > Regarding Q2, I may have overlooked this, but I was unable to find discussion of the history filter in the appendix. Could the authors please clarify where this is addressed?
> >
> > Finally, for Q3, I was unfortunately unable to fully follow the authors’ response. I would encourage the authors to elaborate further on this point in the revised version to improve clarity for readers.

---

> > > ### Author Response · Authors · 2026-04-04
> > >
> > > We thank the reviewer for these further questions.
> > >
> > > ### W1
> > >
> > > Our core contribution is latent group factorization through a VAE–lrRNN. Within this family, two baselines are always present across our experiments: $G=1$ reduces to plain lrRNN, and $G=K$ corresponds to a dimension-wise factorized RNN (finest group splitting in the same model class). Other methods, including bdRNN and SMC are supplementary comparisons included only in specific experimental settings where they add interpretability or context support, rather than as universal counterparts to FacRNN. We agree that consistent baselines matter; in the final paper, we will state explicitly which methods are the fixed core baselines versus ancillary comparison methods.
> > >
> > > ### W5
> > >
> > > This is a very good question. We agree that relying only on the PCA cumulative explained-variance ratio is not sufficient to label a direction as a dummy variable. In our setting, a genuine dummy should behave like (approximately) Gaussian noise; classical ICA arguments (Hyvärinen and Oja, 2000) emphasize that statistical independence among components is informative precisely when sources are non-Gaussian (“independence is non-Gaussian”). Thus, a direction can contribute little incremental variance after PCA yet still carry structured, non-Gaussian information, and should not be dismissed as Gaussian dummy noise on variance alone.
> > >
> > > We therefore supplement the explained-variance curves with normality tests on each within-group PCA coordinate
> > > * Group 0
> > >   * explained variance ratio: `[0.64299828, 0.85100238, 0.99999126, 1.0]`
> > >   * normality test $p$-value: `[4.25e-10, 1.35e-34, 4.36e-17, 0.774]`
> > > * Group 1
> > >   * explained variance ratio: `[0.65260285, 0.99942801, 0.99992943, 1.0]`
> > >   * normality test $p$-value: `[2.25e-08, 2.03e-04, 1.63e-03, 1.87e-02]`
> > >
> > > For Group 1, the third dimension adds only a small increment to the cumulative explained-variance ratio, but its normality test yields $p\approx 1.6\times 10^{-3} < 0.005$, indicating a clear departure from Gaussianity. We therefore do not treat that coordinate as pure Gaussian dummy noise; together with the ground-truth rank $H=3$, this supports retaining three effective non-Gaussian sources per group rather than stopping at two based on the variance elbow alone. We will add this PCA + normality discussion and cite Hyvärinen and Oja (2000) alongside it in the revision.
> > >
> > > ### Q2
> > >
> > > We apologize for the confusion. Across the reviews, we took the message that the higher-order (history-filter) formulation risks feeling redundant relative to what the paper actually uses, and the first-order case ($L=1$, no learned history kernel) in the main text may improve presentation clarity. To clarify, the initial submission has not yet been reorganized that way. If the reviewers agree that this split is preferable, we will move the history-filter generalization to the appendix.
> > >
> > > To answer the original question of Q2: Our formulation includes a history kernel to keep the model general and consistent with neuroscience motivation (e.g., presynaptic effects with temporal decay/refractory-like influence, similar in spirit to GLM-style temporal filters, Pillow et al., 2008). However, in all experiments in this submission, we use the first-order setting ($L=1$). Therefore, no extra temporal-kernel degree of freedom is introduced in the reported results, and the concern about tuning multi-lag kernels does not affect our current empirical claims. For higher-order settings, we fix $\psi$ to a predefined kernel (e.g., exponential decay) rather than learning it, avoiding extra disentanglement degrees of freedom. For clarity, we **will** restrict the main text to the first-order case and move the generalized convolutional formulation to the appendix.
> > >
> > > ### Q3
> > >
> > > We thank the reviewer for raising this again, and we apologize that our earlier explanation was unclear. In the monkey experiment, we do not perform per-latent-dimension analysis or alignment (in the reviewer’s wording, per-"axis" matching of individual latent coordinates). We always align and interpret an entire latent group, which is consistent across the whole paper. As illustrated in Fig. 3(a), second row, we use two latent groups, each with within-group rank $H=2$: one group corresponds to horizontal ($x$) latent dynamics and the other to vertical ($y$) latent dynamics. Biomechanically, a reach in which the hand first accelerates and then decelerates along $x$ is not well described by a 1D dynamical system, so we model it with a 2D latent group and then project that 2D group onto the 1D measured hand $x$ trajectory for comparison—a projection from a group subspace onto a scalar behavioral variable. Throughout, our scientific claims and visualizations are at the group level, not at the level of isolating a single latent dimension as the sole carrier of $x$ or $y$. We will state this distinction explicitly in the revised text and figure captions to avoid axis-wise misreading.

---

### Official Review · Reviewer_qn8d · 2026-03-09

**Soundness:** 3
**Presentation:** 3
**Significance:** 3
**Originality:** 2
**Overall Recommendation:** 5
**Confidence:** 3

**Summary:**

This paper introduces FacRNN (Factored Recurrent Neural Network), a framework for fitting low-rank RNNs to neural population recordings while explicitly enforcing group-wise independence among subsets of latent dimensions. The core idea is to reformulate the standard lrRNN as a variational autoencoder (VAE), which makes it possible to apply a partial correlation (PC) penalty that pushes the aggregated posterior toward a product of group-marginals. Within each group, latent components may remain entangled; across groups, independence is encouraged. A key downstream benefit of this factorization is that the overall connectivity matrix decomposes naturally into interpretable sub-connectivities, each associated with one independent latent group. The framework is evaluated on a synthetic dataset with known ground-truth groups (Lorenz and Thomas attractors), macaque M1 spike train recordings during a center-out reaching task, and mouse dorsal cortex voltage imaging during an air-puff stimulus.

**Compliance With Llm Reviewing Policy:**

Affirmed.

**Final Justification:**

After reading the author rebuttal and the responses to all reviewers, I raise my recommendation to 5: Accept. The authors provided a concrete $\beta$ ablation on the M1 dataset that directly addressed my main open concern. Remaining issues are editorial and the authors have committed to addressing them in the revision.

**Key Questions For Authors:**

1. Is the group decomposition $W = \sum_g W_g$ identifiable under the model, even up to within-group transformations? Specifically, I was not able to see what prevents two different partitions of the latent components from producing equivalent values of the ELBO + PC objective. A brief theoretical discussion, or a reference to a known result, would significantly strengthen confidence in the recovered structure.

2. The ablation for $\beta$ is conducted only on the synthetic dataset where the ground-truth latent $R^2$ can be measured directly. On real data, how should a practitioner choose $\beta$? Is there a principled criterion, such as a held-out likelihood, the elbow in between-group total correlation, or an information criterion? This is a significant practical concern.

3. The PC penalty in Eq. (10) appears to be taken directly from that work. I was not able to identify precisely what FacRNN contributes technically beyond the lrRNN context relative to Li et al. (2025). Could the authors clarify this explicitly, and specify what would be lost if the connectivity estimation component were removed?

4. The M1 dataset has only 14 time bins per trial. I was not able to assess how reliably the history convolution kernel $\psi \in \mathbb{R}^L$ can be estimated in this regime. Do the authors have results suggesting that the learned $\psi$ is stable across random seeds, or that the choice of history length $L$ does not strongly influence the factorization outcome?

**Limitations:**

The authors discuss the need to pre-specify $(G, H)$ and the restriction of interpretable connectivity to linear encoder and decoder settings. However, the practical difficulty of selecting $\beta$ on real data, the absence of any identifiability discussion, and the limited scale of the real-data experiments are not addressed. These should be discussed more explicitly in a revised limitations section.

**Strengths And Weaknesses:**

**Soundness.**

The reformulation of lrRNN as a VAE (Eqs. 4–6) is clean and consistent. The distinction between orthogonality (which SVD enforces) and statistical independence (which the PC penalty targets) is clearly motivated. The block-diagonal RNN (bdRNN) variant is presented honestly as theoretically grounded only for linear dynamics, and the PC-based FacRNN is correctly positioned as the more general approach. The synthetic experiment is informative, and the $\beta$ ablation in Appendix A.2.1 provides useful evidence for the choice $\beta = 20$.

However, I was not able to find a discussion of identifiability of the recovered group structure: it is not clear whether the decomposition $W = \sum_g W_g$ is unique, or whether the PC penalty is sufficient to recover the correct partition in general. I was also not able to identify guidance for selecting $\beta$ on real data where ground truth is unavailable. The ablation is conducted only on the synthetic dataset, and it is unclear how to proceed in practice without access to ground-truth latent $R^2$.

**Presentation.**

The paper is clearly written and easy to follow. The schematic in Figure 1 is helpful. Some points were unclear to me. The term "DisRNN" appears in Figure 4 but is never defined in the text, which I found confusing. The biological interpretations in Section 4.3 (for example, attributing RSC connectivity to episodic memory of the airpuff) are speculative. Framing them as hypotheses rather than conclusions would make the discussion more precise.

**Significance.**

The problem of identifying independent latent subspaces in neural recordings is important and well motivated. The sub-connectivity decomposition in Eq. (12) is elegant and maps directly onto the concept of sub-circuits driving independent computational processes.

The scope of the real-data experiments is limited. The M1 dataset has 168 trials of only 14 time bins each, which is short for a temporal model that estimates a history convolution kernel $\psi$. It is not easy to assess from these experiments whether FacRNN will scale to modern large-scale recordings (thousands of neurons, many time steps per trial). The absence of external task inputs in the real-data experiments is acknowledged by the authors, but it also limits the interpretive conclusions about circuit-level function that can be drawn.

**Originality.**

The VAE reformulation of lrRNN and the PC penalty for group-wise independence are technically sound. However, the PC penalty is imported directly from Li et al. (2025), which is currently an arXiv preprint. The original contribution relative to that work appears to be (a) situating the penalty within the lrRNN connectivity estimation framework and (b) deriving the sub-connectivity decomposition $W = \sum_g W_g$. The authors should more explicitly state what is new relative to Li et al. (2025), particularly since that work may be under concurrent review. The sub-connectivity interpretation and the M1 rotation analysis are meaningful contributions from the perspective of computational neuroscience, but the methodological novelty is more incremental than the framing suggests.

---

> ### Author Rebuttal · Authors · 2026-03-29
>
> Dear Reviewer qn8d,
>
> Thank you very much for your time and valuable comments on our paper. We highly appreciate your recognition of the strengths of our paper. Hopefully, the following responses could resolve most of your concerns and answer your questions.
>
> ### W1 and Q1: Identifiability of the group decomposition $\boldsymbol W = \sum_{g=1}^G \boldsymbol W_g$
>
> We thank the reviewer for raising this. We do not claim strict global identifiability (unique decomposition) under all data-generating settings. Similar to many latent-variable models, group assignments can be non-unique up to permutation and within-group invertible transformations; in weakly separated regimes, multiple near-equivalent optima may exist. In particular, within-group entanglement is invariant up to invertible affine transformations, following the same equivalence argument below Eq. (3): if $(\boldsymbol A,\boldsymbol B)$ is transformed as $(\boldsymbol A\boldsymbol P, \boldsymbol P^{-1}\boldsymbol B)$, one obtains an equivalent latent representation $\boldsymbol z' = \boldsymbol P^{-1}\boldsymbol z$ with unchanged reconstruction/dynamics fit. This is also why in Fig. 7 we evaluate recovery by aligning latent groups to ground-truth groups via affine transformations, rather than by raw axis-wise matching. That said, especially under linear encoder-decoder settings, the group-wise independent latent structure is strongly data-constrained and can be determined in practice through the objective and model-selection pipeline: one can start with conservatively large $G$ and $H_g$, let the model reveal effective group usage/effective ranks, and then refine $(G,H_g)$ based on held-out fit plus independence diagnostics. Therefore, our claim is empirical recoverability under informative structure (validated in synthetic data), not unconditional identifiability. We will revise wording to make this distinction explicit in both the method and the limitations.
>
> ### W2 and Q2: How to choose $\beta$ on real data without ground-truth latent $R^2$?
>
> We agree this is a key practical issue. On real data without latent ground truth, our recommended protocol is to tune $\beta$ using held-out reconstruction/ELBO jointly with a dependence metric (e.g., between-group PC), and select a Pareto point balancing fidelity and factorization (often near an elbow). Similar to most disentanglement methods built on the VAE framework (e.g., $\beta$-VAE, Burgess et al., 2018; $\beta$-TCVAE, Kim and Mnih, 2018; Chen et al., 2018; PDisVAE, Li et al., 2025), $\beta$ is indeed a hyperparameter that currently needs cross-validation, and there is no universally reliable fully automatic rule yet. We agree that this is an important open direction for future work. In practice, existing studies suggest that a moderate range (typically $\beta\in[2,20]$) often works well as a starting point.
>
> ### W3 and Q3: Technical contribution of FacRNN relative to Li et al. (2025)
>
> We appreciate this request for clarification. Our contribution is not proposing a new generic disentanglement penalty, but integrating group-wise partial disentanglement into low-rank recurrent dynamical modeling with explicit connectivity parameterization: (i) VAE reformulation of lrRNN with history-dependent recurrent dynamics; (ii) group-factorized latent dynamics tied to low-rank recurrent weights; (iii) sub-connectivity decomposition $\boldsymbol W=\sum_g\boldsymbol W_g$ that links latent groups to interpretable circuit-level hypotheses. Without connectivity estimation, one may still obtain factorized latent representations, but would lose the sub-circuit interpretation central to neural mechanism analysis, i.e., there will be no neuron-to-neuron interactions that drive the underlying latent dynamics, and hence.
>
> ### W4 and Q4: Reliability of $\psi$ estimation with only 14 time bins per trial (M1 dataset)
>
> See the response to W2 and Q3 in Reviewer Thwq.
>
> ### W5: "DisRNN" undefined in Figure 4
>
> We apologize for this typo. It should be FacRNN.
>
> ### W6: Biological interpretations in Section 4.3 are speculative
>
> We agree. We will revise the text to present these statements as hypotheses consistent with observed connectivity patterns and prior literature, rather than definitive functional claims. We will also make clear that causal validation would require targeted perturbation/recording experiments beyond the scope of this work.
>
> ### Limitation 1: Please revise the limitations section to cover the main practical and conceptual gaps
>
> We agree and will revise the limitations section accordingly. Specifically, we will explicitly enumerate: (i) sensitivity/model selection for $(\beta, G, H)$ on real data; (ii) current real-world experiments do not include explicit task/stimulus regressors, so external vs. internal contributions are not disentangled; (iii) connectivity interpretability is currently limited to linear encoder-decoder settings, while nonlinear variants trade interpretability for flexibility.

---

> > ### Author Rebuttal · Reviewer_qn8d · 2026-04-03
> >
> > I thank the authors for their detailed and constructive rebuttal. Several of my concerns have been addressed. Below I summarize what I consider resolved and what remains open.
> >
> > The proposed protocol (held-out ELBO jointly with between-group PC, selecting a Pareto point near an elbow) is a reasonable practical heuristic. However, I would find it helpful if the authors could show, even briefly, that this protocol leads to a consistent $\beta$ choice on at least one of the real datasets, rather than only on the synthetic one where ground truth is available.
> >
> > Overall, I maintain my recommendation of Weak Accept. The rebuttal has not introduced new concerns, and the planned revisions go in the right direction.

---

> > > ### Author Response · Authors · 2026-04-05
> > >
> > > We thank the reviewer for this suggestion and have added a supplementary ablation on the monkey M1 dataset. We sweep $\beta$ and report held-out reconstruction log-likelihood and between-group partial correlation (PC), each as mean (std) across 5 runs.
> > >
> > > | $\beta$ | Recon log-likelihood | Between-group PC |
> > > |--------:|----------------------|------------------|
> > > | 1 | 380.001 (0.465) | 0.0165 (0.0075) |
> > > | 2 | 379.923 (0.531) | 0.00922 (0.00447) |
> > > | 5 | 379.740 (0.635) | 0.00237 (0.00180) |
> > > | 10 | 379.622 (0.389) | 0.000932 (0.000701) |
> > > | 20 | 379.661 (0.405) | 0.000620 (0.000383) |
> > > | 50 | 379.642 (0.246) | 0.000427 (0.000195) |
> > > | 100 | 379.668 (0.254) | 0.000302 (0.000115) |
> > > | 200 | 379.422 (0.298) | 0.000220 (0.000159) |
> > >
> > > The between-group PC curve shows a clear elbow in the range $\beta \approx 10\sim 20$, after which it largely flattens (changes are small relative to the steep drop at small $\beta$). The reconstruction log-likelihood drops from small $\beta$, is relatively flat through the mid range, and drops again at very large $\beta$ (e.g., 200), matching the expected fidelity–factorization trade-off. Taking both curves together, a balanced operating range is $\beta \approx 10\sim 20$, which aligns with the synthetic ablation in the paper. We will describe this protocol and table in the revised manuscript.

---

### Official Review · Reviewer_q8XB · 2026-03-11

**Soundness:** 3
**Presentation:** 3
**Significance:** 2
**Originality:** 2
**Overall Recommendation:** 5
**Confidence:** 3

**Summary:**

This paper models neural population activity using low-rank RNNs to uncover interpretable latent dynamics and connectivity. The authors propose FacRNN, a factorized low-rank RNN framework that encourages group-wise independent latent dynamics through a VAE formulation with a partial correlation penalty, improving disentanglement and interpretability on both synthetic and real neural datasets.

**Compliance With Llm Reviewing Policy:**

Affirmed.

**Final Justification:**

The authors have mostly addressed my concerns,. Overall, I think the proposed model is reasonable and could be useful, and therefore raised my score to accept.

**Key Questions For Authors:**

- In the real neural datasets, are the learned latent groups related to behavioral variables or brain regions? Additional encoding or region-decoding analyses could help uncover and verify the interpretability of these latent groups.

- Does the low-rank constraint sacrifice predictive performance? For example, could the authors include an ablation study examining how changing the rank affects reconstruction metrics? A brief discussion of what types of neural datasets the proposed method is best suited for would also be helpful for readers.

- How does the model scale to larger neural populations or longer recordings? Clarifying the computational cost would help evaluate the practicality of the method for modern large-scale neural datasets.

**Limitations:**

Yes

**Strengths And Weaknesses:**

Soundness: The approach is technically sound and builds on established low-rank RNN and VAE formulations. The experiments on both synthetic and real neural datasets generally support the claims about improved disentanglement. However, the paper could benefit from stronger quantitative comparisons and additional ablations to clarify the contribution of each component.

Presentation: The paper is well organized and the main idea is understandable. However, more intuitive explanations of the factorization and independence constraints in the method section would help improve clarity.

Significance: Learning interpretable latent dynamics from neural population activity is an important problem in computational neuroscience. While the scope is somewhat specialized, the approach could be useful for researchers studying neural dynamics and low-dimensional structure in large-scale neural trecordings.

Originality: The work proposes combining low-rank RNNs with a factorized latent structure and a partial correlation penalty to encourage independent dynamics across groups. While many components build on existing ideas, this combination for improving interpretability in neural dynamics models is a useful contribution.

---

> ### Author Rebuttal · Authors · 2026-03-29
>
> Dear Reviewer q8XB,
>
> Thank you very much for your time and valuable comments on our paper. We highly appreciate your recognition of the strengths of our paper. Hopefully, the following responses could resolve most of your concerns and answer your questions.
>
> ### W1, W2, and Q1: More analysis and explanations on the factorization and independence. Are the learned latent groups related to behavioral variables or brain regions?
>
> * Thank you for this suggestion. We added a more intuitive interpretation using the monkey M1 analysis. An immediate question is how to interpret the two learned latent groups. In principle, the same hand trajectory can be represented in different coordinate systems (e.g., Cartesian $x, y$ or polar $\rho, \theta$). Empirically, we can stably align the two latent groups to Cartesian horizontal/vertical axes, but not to radius/angle axes (will add the figure to our later revision). **Quantitatively (latent decoding analysis)**, alignment $R^2$ to Cartesian coordinates is about $0.79$, whereas alignment to polar $(\rho,\theta)$ is about $0.24$. This is consistent with prior findings that M1 tuning is not uniform across directions, and is often stronger along cardinal axes (left-right, up-down), potentially reflecting long-term biomechanical and behavioral constraints. Therefore, the recovered latent axes are unlikely to be a purely statistical artifact; they likely capture a meaningful task-relevant subspace in the neural data. This interpretation is also supported by connectivity: instead of only one global low-rank connectivity over all neurons, FacRNN further decomposes it into an $x$ connectivity that encodes horizontal movement and a $y$ connectivity that encodes vertical movement (rather than distance/angle channels), matching the latent-factor interpretation.
>
> * For mouse data, group-specific spatial maps and sub-connectivities align with known cortical ROIs (e.g., S1-bf, M2, S2/RSC pathways), supporting neuroanatomical interpretability. More generally, in unsupervised disentanglement, the primary goal is to let the model automatically discover disentangled latent factors; their post-hoc interpretation can vary by dataset and may correspond to behavior, brain regions, or other latent structure, depending on recording type (single-region neurons vs. whole-brain multi-region signals), scale (small-neuron recordings vs. large-scale imaging pixels), and experimental design.
>
> * We also clarify that we have already conducted substantial ablations on factorization settings: in synthetic data, we provide a dedicated $\beta$ ablation; in both real-world datasets, we compare multiple group specifications (different $(G,H)$ settings) to examine how group structure affects reconstruction and interpretability.
>
> ### Q2: Does the low-rank constraint sacrifice predictive performance? Rank ablation and best-suited datasets?
>
> * Our current results suggest low-rank structure does not necessarily sacrifice predictive quality in these datasets: in synthetic data, methods achieve similar reconstruction while differing strongly in latent disentanglement; in mouse data, reconstruction improves with larger $K$ and plateaus around $K \geqslant 6$, indicating sufficient low-rank capacity can preserve observation fidelity.
> * Importantly, our contribution is not to move from full-rank connectivity/latent to low-rank/low-dimensional modeling per se, but to apply group-wise latent disentanglement. In neural data analysis, low-rank connectivity and low-dimensional latent dynamics are already standard assumptions (within noise tolerance), since the effective latent dimensionality is typically much smaller than the neuron count, and structured/sparse population interactions often induce low-rank organization (e.g., Mastrogiuseppe and Ostojic, 2018; Schuessler et al., 2020a,b; Beiran et al., 2021; Dubreuil et al., 2022; Valente et al., 2022a,b; Pals et al., 2024). Our contribution is to introduce group-wise factorized disentanglement and sub-connectivity interpretability within this established low-rank neuroscience modeling framework.
>
> ### Q3: How does the model scale to larger neural populations or longer recordings?
>
> Thanks for this point. For a sequence with $T$ time points, $G$ groups in $K$ dimensional latent space (group rank $H_g = K/G$ for all groups), the complexity of evaluating the product of marginal groups $q(\boldsymbol z_g),\ g\in\{1,\dots,G\}$ is $\mathcal O(G\times T^2 \times K/G) = \mathcal O(KM^2)$. Specifically, $G$ numbers of $q(\boldsymbol z_g)$. Each one has $T$ points, and each point has a sample, so we are evaluating $T$ samples on $T$ Gaussian kernels. Each Gaussian kernel evaluation is a density of $K/G=H$ dimensions. The number of neurons only affects the VAE's ELBO, so its complexity is the same as plain VAE and independent of the latent disentanglement computation.

---

> > ### Author Rebuttal · Reviewer_q8XB · 2026-04-02
> >
> > The authors have addressed most of my questions, and I would like to raise my score to 5 (accept).

---

> > > ### Author Response · Authors · 2026-04-05
> > >
> > > Thank you for your acknowledgement and for raising the score to 5!

---

### Official Review · Reviewer_Thwq · 2026-03-13

**Soundness:** 3
**Presentation:** 4
**Significance:** 3
**Originality:** 3
**Overall Recommendation:** 5
**Confidence:** 3

**Summary:**

This paper proposes FacRNN to address the lack of interpretability in standard low-rank recurrent neural networks (lrRNNs) applied to high-dimensional neural data. While lrRNNs capture underlying neural structures, their latent dimensions are often entangled, making it difficult to map specific dynamics to computational tasks. The authors frame this as a challenge of group-wise independence. By re-engineering the lrRNN into a variational autoencoder (VAE) and introducing a partial correlation penalty, the model identifies independent latent subspaces. This constraint decouples dynamics between groups while preserving complexity within them, theoretically isolating neural sub-circuits tied to distinct task variables. The authors test the framework on synthetic benchmarks, macaque motor cortex recordings, and mouse voltage imaging. Compared to standard low-rank baselines, FacRNN appears to produce more distinct and interpretable neural trajectories.

**Compliance With Llm Reviewing Policy:**

Affirmed.

**Key Questions For Authors:**

The choice of hyperparameters $G$ and $H$ appears somewhat arbitrary. The authors should clarify how to objectively determine the optimal number of groups and rank in an unsupervised setting—specifically, whether cross-validated ELBO serves as a reliable metric for this selection.

The treatment of external inputs also requires further detail. It remains unclear how the framework explicitly separates task-driven external signals from the internally generated dynamics within the latent groups. Without a clear mechanism for this distinction, the identified sub-circuits may conflate exogenous and endogenous activity.

Regarding the history length $L$, the paper lacks an analysis of its impact on the stability of learned connectivity. I am concerned that improper calibration of $L$ might lead to information "bleeding" between groups, which would undermine the claim of disentanglement.

Finally, while the MLP variant improves performance, it sacrifices the direct interpretability of the linear model. The authors should discuss whether "effective connectivity" could be recovered—perhaps via Jacobian analysis—to maintain circuit-level insights despite the non-linearity.

**Limitations:**

yes

**Strengths And Weaknesses:**

**Strengths**

The work reformulates low-rank RNN dynamics into a discrete VAE framework, which allows for optimization via the ELBO. This shift is theoretically sound. By adopting a group-wise independence assumption rather than the strict independence typical of ICA-based methods, the model achieves better alignment with biological reality. The empirical evaluation is broad, covering synthetic data, macaque electrophysiology, and mouse voltage imaging. Furthermore, the paper is well-organized, and the visualizations of functional sub-circuits effectively demonstrate the model's utility.

**Weaknesses**

Several technical gaps limit the current presentation. The model appears highly sensitive to hyperparameters, specifically the penalty term $\beta$ and the group count $G$. The authors provide little guidance on how to tune these parameters, which is a significant hurdle for practical application. Regarding the architecture, the influence of history length $L$ remains unexamined, and the framework currently lacks a clear mechanism to handle external task inputs. This is a notable omission for a model intended to capture task-related neural dynamics.

There is also a tension between performance and the paper's stated goal of interpretability. The nonlinear FacRNN-MLP variant performs better but obscures the connectivity matrices, yet this trade-off is not discussed in depth. Finally, the mathematical notation is unnecessarily dense in several sections. Important evidence from ablation studies is relegated to the appendix, which weakens the main text's arguments regarding model configuration.

---

> ### Author Rebuttal · Authors · 2026-03-29
>
> Dear Reviewer THwq,
>
> Thank you very much for your time and valuable comments on our paper. We highly appreciate your recognition of the strengths of our paper. Hopefully, the following responses could resolve most of your concerns and answer your questions.
>
> ### W1 and Q1: How to objectively select $\beta$, $G$, and $H$ in an unsupervised setting?
>
> * On real data without latent ground truth, our recommended protocol is to tune $\beta$ using held-out reconstruction/ELBO jointly with a dependence metric (e.g., between-group PC), and select a Pareto point balancing fidelity and factorization (often near an elbow). Similar to most disentanglement methods built on the VAE framework (e.g., $\beta$-VAE, Burgess et al., 2018; $\beta$-TCVAE, Kim and Mnih, 2018; Chen et al., 2018; PDisVAE, Li et al., 2025), $\beta$ is indeed a hyperparameter that currently needs cross-validation, and there is no universally reliable fully automatic rule yet. We agree that this is an important open direction for future work. In practice, existing studies suggest that a moderate range (typically $\beta\in[2,20]$) often works well as a starting point.
> * On selecting $G$ and $H$: using PC to enforce group-wise independence is more flexible than fully factored settings, because we do not force each latent dim to be independent. Instead, we assume independence across multi-dimensional groups while allowing within-group entanglement. When the true group structure is unknown, we can conservatively set a sufficiently large number of groups $G$ and sufficiently large within-group ranks $H_g$, then let the model reveal effective group usage. When $H_{\mathrm{true}}$ is unknown, the extra dims in each group naturally act as dummy variables after effective-rank identification (see W5, Reviewer 7MBW). In practice, one can start with conservatively large $G$ and $H_g$, inspect the effective rank of each group, and then rerun with refined $(G, H_g)$ for cleaner results. For real neural data, this model-selection process can be further informed by domain priors and neuroscience interpretations.
>
> ### W2 and Q3: Impact of history length $L$ on disentanglement stability
>
> Thank you for this important point. Our formulation includes a history kernel to keep the model general and consistent with neuroscience motivation (e.g., presynaptic effects with temporal decay/refractory-like influence, similar in spirit to GLM-style temporal filters, Pillow et al., 2008). However, in all experiments in this submission, we use the first-order setting ($L=1$). Therefore, no extra temporal-kernel degree of freedom is introduced in the reported results, and the concern about tuning multi-lag kernels does not affect our current empirical claims. For higher-order settings, we fix $\psi$ to a predefined kernel (e.g., exponential decay) rather than learning it, avoiding extra disentanglement degrees of freedom. For clarity, we restrict the main text to the first-order case and move the generalized convolutional formulation to the appendix.
>
> ### W3 and Q2: How does the framework separate task-driven external signals from internally generated dynamics?
>
> Thank you for this point. Our framework explicitly models external input and separates it from recurrent internal dynamics in the generative equations (Eq. 1,2,5,7,7,8,11). In the synthetic experiment (Sec. 4.1, Dataset), external input is actually included in data generation and is treated as known, so it does not confound the disentanglement analysis there. For the real datasets used in this paper, the true external input is not directly observed/annotated, so we did not instantiate an explicit input regressor in those specific experiments. More generally, if a dataset provides external covariates $\phi^{(t)}$, they can be incorporated through a learnable linear projection (external-dim $\to N$ neurons), added as an explicit exogenous-drive term. Under this formulation, the model can distinguish external input contributions from internally generated recurrent dynamics.
>
> ### W4 and Q4: Interpretability vs. performance trade-off in the FacRNN-MLP variant
>
> Thank you for this very helpful suggestion. We agree that increased neural-network flexibility can affect latent disentanglement behavior, even if it may improve representation capacity. The purpose of introducing FacRNN-MLP is to test a conceptual question: whether increasing encoder/decoder expressiveness alone can substitute for explicit group-structured inductive bias in latent factorization. In other words, FacRNN-MLP is included to separate the role of architecture flexibility from the role of factorization assumptions, rather than to serve as the primary interpretable model. For nonlinear encoder-decoder settings, effective-connectivity proxies can be derived using Jacobian-based local linearization, and potentially complemented by attribution-style tools such as saliency maps and input×gradient analyses. We will add this discussion to the appendix.

---

### Decision · Program_Chairs · 2026-04-30

**Decision:**

Accept (spotlight)

**Comment:**

This work combines two lines of work in modeling of neural dynamics, variational autoencoders and low-rank recurrent neural networks, by reformulating the latter in terms of the former. This allows the authors to propose a network connectivity structure that factorizes latent dimensions in terms of low-rank subspaces that mix under dynamics but are largely independent of each other. They apply this method to several neuroscience, arguing for interpretability of the findings.

Reviewers agreed about the significance of the problem addressed for systems and computational neuroscience, as well as the value of casting low-rank neural networks into the VAE approach via a regularization of partial correlation. In addition, reviewers appreciated the breadth of empirical tests of the model. However, some reviewers noted inconsistencies in the settings across experiments, and one reviewer noted that the model did not automatically determine the number of low-rank subspaces as claimed.

Overall, a valuable contribution to efforts linking machine learning to neuroscience via interpretable dynamics.